# Developmentally regulated *Tcf7l2* splice variants mediate transcriptional repressor functions during eye formation

**Rodrigo M Young[1][†]\*, Kenneth B Ewan[2], Veronica P Ferrer[1], Miguel L Allende[3], Jasminka Godovac-Zimmermann[4], Trevor C Dale[2], Stephen W Wilson[1]\***

[1]Department of Cell and Developmental Biology, UCL, London, United Kingdom; [2]School of Bioscience, Cardiff University, Cardiff, United Kingdom; [3]FONDAP Center for Genome Regulation, Facultad de Ciencias, Universidad de Chile, Santiago, Chile; [4]Division of Medicine, UCL, London, United Kingdom

**Abstract** Tcf7l2 mediates Wnt/β-Catenin signalling during development and is implicated in cancer and type-2 diabetes. The mechanisms by which Tcf7l2 and Wnt/β-Catenin signalling elicit such a diversity of biological outcomes are poorly understood. Here, we study the function of zebrafish *tcf7l2* alternative splice variants and show that only variants that include exon five or an analogous human *tcf7l2* variant can effectively provide compensatory repressor function to restore eye formation in embryos lacking *tcf7l1a/tcf7l1b* function. Knockdown of exon five specific *tcf7l2* variants in *tcf7l1a* mutants also compromises eye formation, and these variants can effectively repress Wnt pathway activity in reporter assays using Wnt target gene promoters. We show that the repressive activities of exon5-coded variants are likely explained by their interaction with Tle co-repressors. Furthermore, phosphorylated residues in Tcf7l2 coded exon5 facilitate repressor activity. Our studies suggest that developmentally regulated splicing of *tcf7l2* can influence the transcriptional output of the Wnt pathway.

**\*For correspondence:**
rodrigo.young@ucl.ac.uk (RMY); s.wilson@ucl.ac.uk (SWW)

**Present address:** [†]Institute of Ophthalmology, UCL, London, United Kingdom

**Competing interests:** The authors declare that no competing interests exist.

## Introduction

Wnt signalling has a broad array of biological functions, from regional patterning and fate specification during embryonic development to tissue homeostasis and stem cell niche maintenance in adult organs (*van Amerongen and Nusse, 2009*; *Nusse and Clevers, 2017*). Because of its relevance to such a diversity of processes, Wnt pathway misregulation is linked to a range of diseases such as cancer and diabetes, and neurological/behavioural conditions (*Nusse and Clevers, 2017*). Wnts can activate several intracellular pathways, and the branch that controls gene expression works specifically through β-catenin and the small family of T-Cell transcription factors (Tcfs; *Cadigan and Waterman, 2012b*).

   In absence of Wnt ligand, intracellular β-catenin levels are kept low by a mechanism that involves its phosphorylation by GSK-3β and CK1α, which is mediated by the scaffolding of β-catenin by Axin1 and APC, in what is termed the destruction complex (*MacDonald and He, 2012*; *Niehrs, 2012*). Phosphorylated β-catenin is ubiquitilated and degraded in the proteasome (*MacDonald et al., 2009*; *Niehrs, 2012*). In this context, Tcf transcription factors actively repress the transcription of downstream genes by interacting with Groucho(Gro)/TLE co-repressors (*Cadigan and Waterman, 2012b*; *Hoppler and Waterman, 2014*). When cells are exposed to Wnt ligand, the destruction complex is disassembled and β-catenin is no longer phosphorylated and committed to degradation (*MacDonald et al., 2009*; *Niehrs, 2012*). This promotes the translocation of β-catenin to the nucleus where it displaces Gro/TLE co-repressors from interacting with Tcfs and activating the transcription of Wnt target genes (*Cadigan and Waterman, 2012b*; *Hoppler and Waterman, 2014*;

Schuijers et al., 2014). Hence, Tcf proteins are thought to work as transcriptional switches that can activate transcription in the presence of Wnt ligands or repress transcription in their absence.

During development, ensuring appropriate levels of Wnt/β-catenin signalling is essential for many processes. For instance, during gastrulation, specification of the eyes and telencephalon can only occur when Wnt/β-catenin signalling is low or absent, and overactivation of the pathway in the anterior neuroectoderm mispatterns the neural plate leading to embryos with no eyes (Kim et al., 2000; Heisenberg et al., 2001; Kiecker and Niehrs, 2001; van de Water et al., 2001; Houart et al., 2002; Dorsky et al., 2003). Illustrating this, fish embryos mutant for axin1, a member of the β-catenin destruction complex, are eyeless because cells fail to phosphorylate β-catenin, leading to abnormally high levels of the protein, mimicking a Wnt active state (Heisenberg et al., 2001; van de Water et al., 2001). Similarly, it has been shown that tcf7l1a/headless mutants also mimic Wnt/β-catenin overactivation suggesting that it is necessary to actively repress Wnt/β-catenin target genes for regional patterning to occur normally (Kim et al., 2000; Young et al., 2019).

In vertebrates, Lef/Tcf transcription factors constitute a family of four genes: lef1, tcf7 (tcf1), tcf7l1 (tcf3) and tcf7l2 (tcf4). All contain a highly conserved β-catenin binding domain (β-catenin-BD) at the amino-terminal (N-terminal) end and a high mobility group box (HMG-box) DNA binding domain in the middle of the protein (Figure 1A; Cadigan and Waterman, 2012b; Hoppler and Waterman, 2014). All Lef/Tcf proteins bind the 5′-CCTTTGATS-3′ (S = G/C) DNA motif, but can also bind to sequences that diverge from this consensus (van de Wetering et al., 1997; van Beest et al., 2000; Hallikas and Taipale, 2006; Atcha et al., 2007). The fact that all Lef/Tcfs bind to the same motif has led to the notion that the functional specificity of Lef/Tcf proteins may be imparted by inclusion or exclusion of functional motifs by alternative transcription start sites or alternative splicing (Hoppler and Waterman, 2014).

The region between the β-catenin-BD and the DNA binding domain of Lef/Tcf proteins, known as the context-dependent regulatory domain (CDRD), and the carboxy-terminal (C-terminal) end of the protein, are coded by alternatively spliced exons (Figure 1A; Young et al., 2002; Archbold et al., 2012; Cadigan and Waterman, 2012b; Hoppler and Waterman, 2014). The C-terminal region of Lef/Tcfs includes the C-Clamp domain and two CtBP interacting motifs (Figure 1A; Brannon et al., 1999; Valenta et al., 2003; Atcha et al., 2007; Hoverter et al., 2012; Hoverter et al., 2014). In certain contexts, the C-clamp domain helps DNA binding and increases the selectivity for certain gene promoters (Atcha et al., 2007; Wöhrle et al., 2007; Chang et al., 2008; Weise et al., 2010). The CDRD includes the domain of interaction with Gro/TLE co-repressors (Groucho Binding Site (GBS), Figure 1A; Cavallo et al., 1998; Roose et al., 1998; Daniels and Weis, 2005; Arce et al., 2009; Chodaparambil et al., 2014) and amino acids that can promote the dissociation of Lef/Tcfs from DNA, modify nuclear localisation or promote activation of transcription when phosphorylated by HIPK2, TNIK or NLK (Shetty et al., 2005; Mahmoudi et al., 2009; Hikasa et al., 2010; Ota et al., 2012). Exons 4 and 5 and the borders of exon7 and exon 9, which are included in the region of the CDRD, are alternatively spliced in tcf7l2 (Duval et al., 2000; Pukrop et al., 2001; Young et al., 2002). The inclusion of the border of exon nine can transform Tcf7l2 into a strong transcriptional repressor (Liu et al., 2005). Hence, splicing regulation in the CDRD could, similarly, be relevant to transcriptional output (Tsedensodnom et al., 2011; Koga et al., 2012). However, the function of the alternatively spliced exons 4 and 5 is still unknown. All the variations in Lef/Tcf proteins described above may contribute to their functional diversity, an idea that is supported by the fact that Lef/Tcfs control many different subsets of genes (Cadigan and Waterman, 2012b; Hrckulak et al., 2016).

Tcf7l2 has various known roles including a requirement during establishment of left-right asymmetry in the habenulae and for the maintenance of the stem cell compartment in colon and skin epithelia (Korinek et al., 1998; Nguyen et al., 2009; Hüsken et al., 2014). Additionally, polymorphisms located in introns in the genomic region that codes for human tcf7l2 exon 3a (Figure 1 C, D), segregate with acquisition of type-2 diabetes (Grant et al., 2006), and conditional knockdowns of tcf7l2 give rise to mice with phenotypes comparable to diabetic patients (Boj et al., 2012; Duncan et al., 2019). tcf7l2 has an alternative translation start site and alternative splicing in the CDRD and in exons that lead to shorter C-terminal ends (Duval et al., 2000; Young et al., 2002; Vacik et al., 2011). Overall, this suggests that many regulatory inputs could influence the transcriptional output of Tcf7l2.

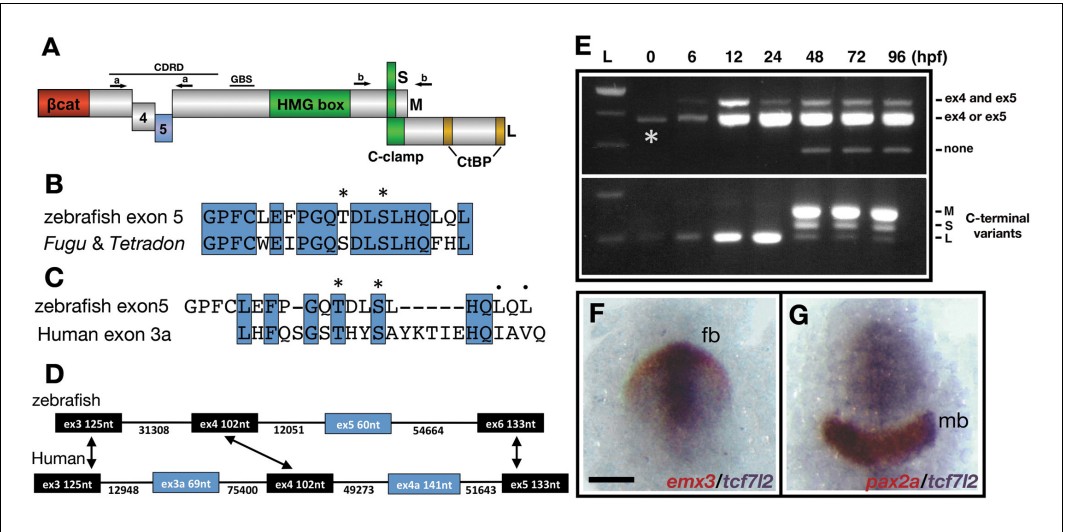

**Figure 1.** Description and expression of a new alternatively spliced exon in zebrafish *tcf7l2*. (**A**) Schematic representation of variants of Tcf7l2 arising from different splice forms (not to scale). Labels 4 and 5 represent the region of Tcf7l2 coded by alternative exons 4 and 5. Short (S) Medium (M) and Long (L) C-terminal variants coded by alternative splice variants in the 5' end of exon 15 are indicated. Red box, β-catenin (βcat) binding domain. Green boxes, High-Mobility Group (HMG) Box, which is the primary DNA interacting domain, and C-clamp DNA-helper binding domain. Yellow boxes, CtBP interaction domains. CDRD labelled line over exons 4 and 5 indicates the Context Dependent Regulatory Domain and Groucho Binding Site (GBS) marks the region of interaction with Groucho/Tle transcriptional co-repressors. Arrows indicate the position of primer sets 'a' and 'b' used for RT-PCR experiments in (**E**). (**B–C**) Alignment of the amino acid sequences coded by zebrafish, *Takifugu rubripens* and *Tetradon tcf7l2* exon 5 (**B**) or human exon 3a (**C**). Identical amino acids marked by blue boxes. Asterisks over sequence mark putative phosphorylated amino acids. Dots over sequence indicate similar amino acids. (**D**) Schematic of the genomic region of zebrafish and human *tcf7l2*. Introns depicted as lines and exons as boxes. Blue exon boxes depict human *tcf7l2* alternative exons 3a and 4a, and zebrafish alternative exon 5. Black exon boxes indicate equivalent exons in both species emphasised by arrows. Numbers under introns and within exons represent their nucleotide size (not to scale). (**E**) RT-PCR experiments performed on cDNA from embryos at stages indicated in hours post fertilisation (hpf). L, 1 Kb ladder. Top panel shows results of PCRs using primer set 'a' (indicated in *Figure 1A*, Materials and methods) amplifying the region of alternative exons 4 and 5. Middle band contains amplicons including either *tcf7l2* exon 4 or exon 5. Bottom panel shows results of PCRs using primer set 'b' (indicated in *Figure 1A*, Materials and methods) amplifying the region of alternative exon 15. Asterisk shows maternal expression of *tcf7l2*. (**F–G**) Double in situ hybridisation of *tcf7l2*, in blue, and *emx3* (**F**) or *pax2a* (**G**) in red. 10hpf flat mounted embryos, dorsal view, anterior up, posterior down; fb, prospective forebrain; mb, prospective midbrain. Scale Bar in (**F**) is 200 μm.

The online version of this article includes the following source data and figure supplement(s) for figure 1:

**Source data 1.** Zebrafish exon five nucleotide and coded amino acid sequences.
**Figure supplement 1.** Alignment of the amino acid sequence coded by *tcf7l2* exon five in zebrafish and other fish species.

In this study, we address the role of alternative splicing in mediating the functional properties of Tcf7l2 during early nervous system development. Our results show that alternative splicing of *tcf7l2* significantly impacts the transcriptional repressor activity of the encoded protein. *tcf7l2* splice variants have been characterised in humans and, to a lesser extent, in mice and zebrafish (*Duval et al., 2000*; *Young et al., 2002*; *Prokunina-Olsson et al., 2009*; *Weise et al., 2010*), but little information is available on different roles for the splice variants. In zebrafish, *tcf7l2* is first expressed in the anterior neuroectoderm by the end of gastrulation (*Young et al., 2002*) and in this study, we show that at this stage, *tcf7l2* is only expressed as long C-terminal variants that can include a newly identified alternative exon 5. We show that only zebrafish Tcf7l2 variants that include the coded exon five and comparable human Tcf7l2 variants effectively provide the repressive function required to promote eye specification. Moreover, only these variants are effective in repressing Wnt target gene promoters in luciferase assays, probably due to interaction with Gro/Tle co-repressors. We further show

that two phosphorylated amino acids coded by exon 5 of Tcf7l2 are required for this interaction, and overall repressive function. Hence, our results suggest that alternative exon five in zebrafish *tcf7l2* could play a critical role in mediating transcriptional repression of Wnt pathway target genes. Our data also suggest that through inclusion of the region coded by exon 5, Tcf7l2 could be part of a phosphorylation regulatory module that keeps the Wnt pathway in an 'off' state by phosphorylating β-catenin in the cytoplasm and Tcf7l2 in the nucleus.

## Results

### Characterisation of a novel *tcf7l2* alternative splice variant

With the aim of addressing the functional roles of different *tcf7l2* splice variants, we cloned the zebrafish splice forms in the CDRD (*Figure 1A*). This region of Tcf proteins is close to the fragment that interacts with Gro/TLE co-repressors (GBS, *Figure 1A*) and consequently alternative splice forms may affect transcriptional function by modulating interactions with Gro/TLE proteins (*Duval et al., 2000*; *Young et al., 2002*; *Roose et al., 1998*; *Daniels and Weis, 2005*). Using primers flanking the region containing putative alternative exons in the CDRD encoding region (Primer set-a, *Figure 1A*), we performed RT-PCR and cloned the resulting DNA fragments.

The amplified DNA contained a new exon not previously described in zebrafish or in any other species (*Figure 1B*, *Figure 1—source data 1*, accession number MN646677). This exon (*tcf7l2* exon five hereafter) codes for a 20 amino acid stretch and is flanked by consensus splice acceptor and donor intron sequences. This region of human *tcf7l2* also includes the alternatively spliced exons 3a and 4a (*Figure 1D*; *Prokunina-Olsson et al., 2009*). Zebrafish *tcf7l2* exon five is similar in size to human *tcf7l2* exon 3a but instead, lies in a genomic location that in the human gene would be positioned between exons 4 and 5, where alternative human exon 4a is located (*Figure 1D*). Although the protein sequence homology with other fish species is high (*Figure 1B*, *Figure 1—figure supplement 1A*), the amino acid identity encoded by human exon 3a and zebrafish exon 5 is only 33% (*Figure 1C*). However, in both species, all neighbouring exons are the same size, show a high-degree of nucleotide and protein sequence homology, and are surrounded by long introns (*Figure 1D*, *Figure 1—figure supplement 1B*). Moreover, both fish and human exons 5/3a encode residues that are putative CK1/PKA kinase phosphorylation sites (*Figure 1B,C*, asterisks).

RT-PCR experiments showed that splice variants that include alternative exon four are expressed maternally and zygotically (*Figure 1E*, upper panel, middle band labelled '4 or 5', asterisk). Exon five is expressed zygotically and is included in *tcf7l2* transcripts from six hpf (hours post fertilisation) onwards (*Figure 1E*, upper panel, top band labelled '4 and 5'). *tcf7l2* splice variants that lack both exons 4 and 5 are only expressed from 48hpf onwards (*Figure 1E*, upper panel, lower band labelled 'none').

The inclusion of two alternative forms of exon 15 add a premature stop codon that leads to medium and short Tcf7l2 Ct variants (refered as exon 17 in *Young et al., 2002*). We characterised this region by analysing the alternatively spliced 5' end of *tcf7l2* exon 15 for the presence of long, medium or short splice variant C-terminal (Ct) coding ends (Primer set-b, *Figure 1A,E*, lower panel). Maternally, and until 24hpf, *tcf7l2* is only expressed as transcripts that lead to long (L) Tcf7l2 variants (*Figure 1E*, bottom panel, lower band). From 48hpf onwards *tcf7l2* is predominantly expressed as splice forms that code for medium (M) and short (S) Ct Tc7f7l2 variants, with L variants barely detectable (*Figure 1E*, bottom panel, top two bands).

We further assessed how the expression of exons 4 and 5 relate to alternative splice forms in exon 15 affecting the Ct domain of Tcf7l2 (*Figure 2A*). Before 48hpf, *tcf7l2* is only expressed as splice forms that lead to long Ct variants, but from 48hpf, variants including exon four are expressed predominantly as medium variants, but also as short and long Ct-forms (*Figure 2B-C*). On the other hand, at 48hpf and onwards, transcripts including exon five are expressed only as medium or short Tcf7l2 Ct-variants and by 96hpf only as M variants (*Figure 2C*). The range of Tcf7l2 variants expressed as development proceeds is summarised in *Supplementary file 1A-a* and *Supplementary file 1A-b*.

Given the functional relevance of Tcf7l2 in adult tissue homeostasis (*Nusse and Clevers, 2017*), we studied splicing events involving exons 4/5 and exon 15 in adult zebrafish eyes, brain, gut, liver,

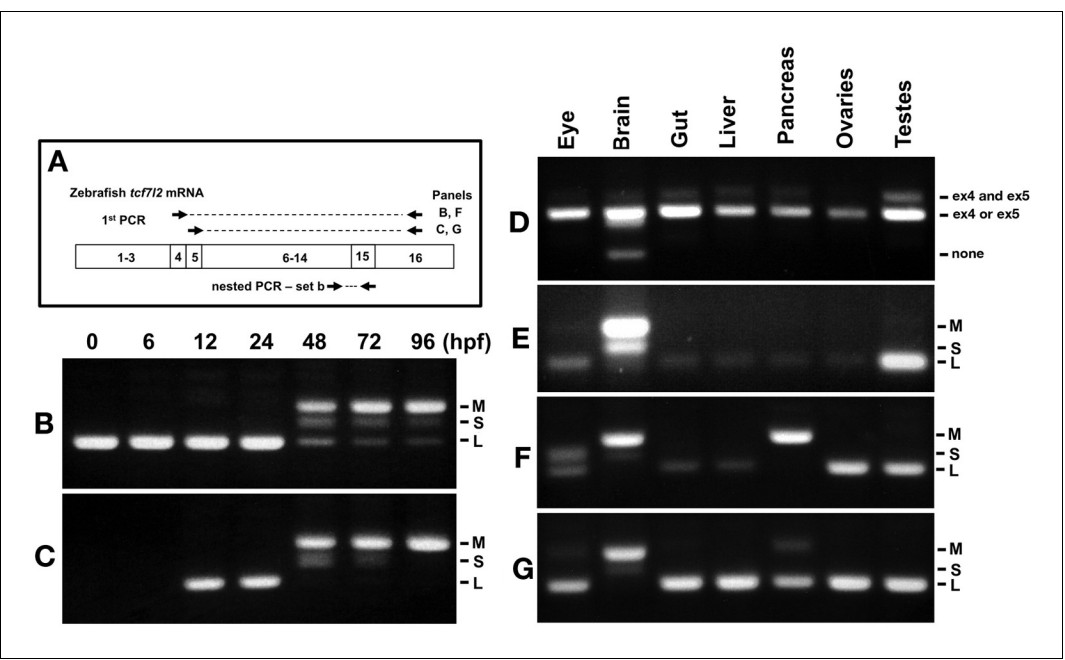

**Figure 2.** Expression of *tcf7l2* alternative exons 4, 5 and 15 varies across development and in adult organs. RT-PCR analysis of alternative exons 4, 5 and 15 of zebrafish *tcf7l2* across development and in various adult organs. (A) Schematic representation of nested RT-PCR strategy used for panels B, C, F and G. (**B–C**) cDNA from embryos of ages indicated (hpf) was PCR amplified using a forward primer that anneals over exon 4 (**B**) or exon 5 (**C**) and a reverse primer that anneals to exon 16 common to all *tcf7l2* mRNA variants. The product from this first PCR was then used as a template for a nested PCR using primer set 'b' (as in *Figure 1*. **A**) that amplifies exon 15 and reveals the different possible Ct ends of Tcf7l2. This last PCR product is shown in these panels. M (Medium), S (short) and L (Long) C-terminal Tcf7l2 variant end. (**D–E**) RT-PCR experiments performed on cDNA of the indicated adult organs using primer set 'a' (Materials and methods) amplifying the region of alternative exons 4 and 5 (**D**) or using primer set 'b' (Materials and methods) amplifying the region of alternative exon 15 (**E**). (**F–G**) Same PCR amplification strategy used in panels (**B–C**) to detect the C-terminal Tcf7l2 variants associated with exons 4 or 5, but using the indicated adult organ cDNA as template in the 1st PCR reaction.

pancreas, ovaries and testis (*Figure 2D-G*). A summary of the RT-PCR results and Tcf7l2 variants expressed in adult organs based on this RT-PCR data is presented in *Supplementary file 1B*.

From here, we focus exclusively on *tcf7l2* splice variants expressed as the anterior neuroectoderm is patterned during gastrula and early somite stages (6-12hpf).

## *tcf7l2* is broadly expressed in the anterior neural plate

The analyses above show that by early somite stage (12hpf), *tcf7l2* is predominantly expressed as two long C-terminal isoforms, both which include exon 4. One that lacks exon 5 (*4L-tcf7l2* variant from here onwards), which is expressed maternally and zygotically, and the other that includes exon five which is expressed zygotically (*45L-tcf7l2* variants from here onwards).

From late gastrula stage, *tcf7l2* is expressed in the anterior neural plate (*Young et al., 2002*), overlapping with *tcf7l1a* and *tcf7l1b* (*Kim et al., 2000*; *Dorsky et al., 2003*). At 10hpf, by the end of gastrulation, the expression of *tcf7l2* overlaps rostrally with that of *emx3* delimiting the prospective telencephalon (*Figure 1F*; *Morita et al., 1995*). The caudal expression of *tcf7l2* in this region ends a few cell diameters anterior to the rostral limit of midbrain marker *pax2a*, suggesting it may extend beyond the diencephalon and include part of the prospective midbrain region (*Figure 1G*). Consequently, *tcf7l2* is expressed throughout the rostral neural plate including the eye field during the stages when the neural plate becomes regionalised into discrete domains.

# Zebrafish *tcf7l2* exon five and human *tcf7l2* exon 3a containing variants are able to restore eye formation upon loss of *tcf7l1a/b* function

Zygotic *tcf7l1a/headless*$^{m881/m881}$ mutants (Z*tcf7l1a*$^{-/-}$ from here onwards) show reduced eye size at 30hpf but later, growth compensation restores eye size (*Figure 3G*; *Figure 1—source data 1*; *Young et al., 2019*). However, Z*tcf7l1a* mutants are sensitised to further loss of Tcf repressor activity and when *tcf7l1b*, the paralogue of *tcf7l1a*, is knocked down by injecting 0.12pmol of a validated ATG morpholino (mo$^{tcf7l1b}$ *Dorsky et al., 2003*), the eye field is not specified (*Figure 3D*, *Figure 1—source data 1*, $\bar{x}$=99%, n=270, 3 experiments; *Dorsky et al., 2003*). To address whether there are any differences in the functional properties of *tcf7l2* splice variants expressed at 12hpf (*Figure 1E*, *Supplementary file 1A-b*, we assessed their ability to restore eye formation in Tcf7l1a/Tcf7l1b abrogated embryos (*Figure 3D*). Z*tcf7l1a*$^{-/-}$ embryos were co-injected with 0.12pmol of mo$^{tcf7l1b}$ and 20pg of either *4L-tcf7l2* or *45L-tcf7l2* mRNA. Control overexpression of these *tcf7l2* variants in wild-type or Z*tcf7l1a*$^{+/-}$ embryos did not induce any overt phenotype (not shown).

Exogenous *45L-tcf7l2* mRNA effectively rescued the eyeless phenotype in Z*tcf7l1a*$^{-/-}$/*tcf7l1b* morphant embryos (*Figure 3E, H*, *Supplementary file 1E*, $\bar{x}$=89.1±3.4%, n=163 embryos, 3 experiments), whereas *4L-tcf7l2* mRNA did not (*Figure 3H*, *Supplementary file 1E*, $\bar{x}$=18.4±4.4%, n=146 embryos, 3 experiments). This suggests that only *tcf7l2* splice variants that include exon 5 have the ability to repress the aberrant pathway activation upon loss of Tcf7l1a/Tcf7l1b function.

An alternative explanation for the poor restoration of eye formation by the 4L-Tcf7l2 variant could be that it is either not localised to the nucleus or has lower protein stability. To address this, we transfected HEK293 cells with constructs encoding N-terminal Myc tagged (MT) constructs of Tcf7l2. Both MT-Tcf7l2 variants were expressed and localised to the nucleus (*Figure 3—figure supplement 1*), suggesting that the lack of rescue with *4L-tcf7l2* mRNA is due to other differences in protein function. The MT-45L-Tcf7l2, but not MT-4L-Tcf7l2 variant was also able to restore eye formation in Z*tcf7l1a*$^{-/-}$/*tcf7l1b* morphants to a similar extent as the untagged variant (*Figure 3H*, *Supplementary file 1E*, $\bar{x}$=70.9±3.9%, n=199 embryos, 3 experiments).

These results suggest that 45-Tcf7l2 but not 4L-Tcf7l2 variant can compensate for loss of Tcf7l1a/b function in eye specification.

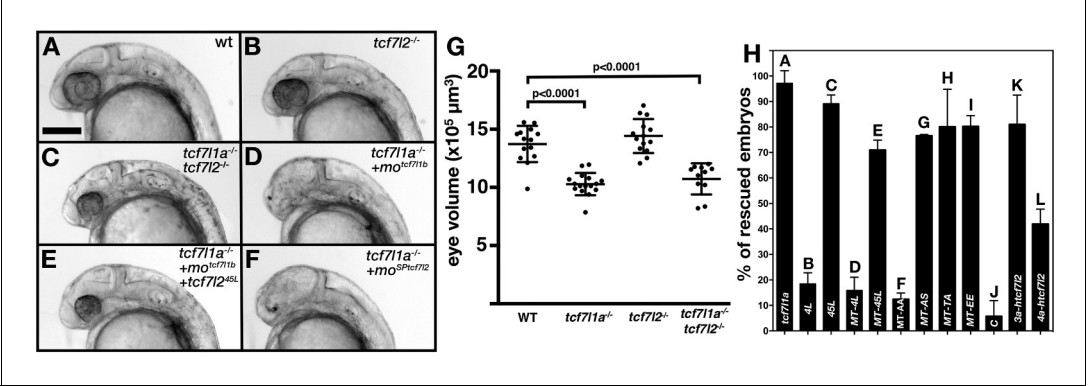

**Figure 3.** Alternative exon 5 of *tcf7l2* impacts eye formation. (A–F) Lateral views (anterior to left, dorsal up) of 28hpf live wildtype (A), *tcf7l2*$^{zf55/zf55}$ (B), double *tcf7l1a*$^{-/-}$/*tcf7l2*$^{zf55/zf55}$ (C) and *tcf7l1a*$^{-/-}$ (D–F) zebrafish embryos with injected reagents indicated top right showing representative phenotypes. (D) 0.12 pmol mo$^{tcf7l1b}$ (E), 0.12 pmol mo$^{tcf7l1b}$ and 20 pg of *45L-tcf72* splice variant mRNA, (F) 1.25 pmol mo$^{SPtcf7l2}$. Scale bar in (A) is 200 µm. (G) Plot showing the volume of eyes (µm$^3$) of 30hpf fixed embryos coming from a double heterozygous *tcf7l1a/tcf7l2* mutant incross. Error bars are mean ± SD, only P values greater than 0.1 from unpaired t test with Welch's correction are indicated. Data in *Supplementary file 1C*. (H) Bars represent the percentage of *tcf7l1a*$^{-/-}$ embryos that develop eyes (with distinguishable lens and pigmented retina) coming from multiple *tcf7l1a*$^{+/-}$ female to *tcf7l1a*$^{-/-}$ males crosses, injected with 0.12 pmol of mo$^{tcf7l1b}$ (all bars) and co-injected with constructs stated on X axis: 10 pg of *tcf7l1a* mRNA (A) 20 pg of *tcf7l2* mRNA splice variants *4L-tcf7l2* (B) *45L-tcf7l2*, (C) *MT-4L-tcf7l2* (D) *MT-45L-tcf7l2* (E) MT-*tcf7l2*-AA (F) MT-*tcf7l2*-AS (G) MT-*tcf7l2*-TA (H) MT-*tcf7l2*-AA (I) *htcf7l2*-C (J) *htcf7l2*-3a (K) and *htcf7l2*-4a (L). Data for all these plots are included in *Supplementary file 1E*. Error bars are mean ± SD.

The online version of this article includes the following figure supplement(s) for figure 3:

**Figure supplement 1.** Tcf7l2 variants localise to the nucleus.

**Figure supplement 2.** Splicing specific morpholino knockdown of *tcf7l2* splice variants that include exon5.

To assess the function of human *tcf7l2* alternative exons in the region of zebrafish *tcf7l2* exon 5, we cloned long Ct human *tcf7l2* splice variant cDNA including alternative exon 3a (*3a-htcf7l2*) or 4a (*4a-htcf7l2*), or excluding both exons (*C-htcf7l2*). *3a-htcf7l2* mRNA expression was able to restore eye formation in most *Ztcf7l1a$^{-/-}$/tcf7l1b* morphant embryos (*Figure 3H*, *Supplementary file 1E*, $\bar{x}$=80.9%±11.6, n=129 embryos, 3 experiments). However, *4a-htcf7l2* mRNA restored eye formation less effectively (*Figure 3E*, *Supplementary file 1E*, $\bar{x}$=41.9±5.8%, n=197 embryos, 3 experiments), and the variant lacking both alternative exons failed to restore eye formation (*Figure 3H*, *Supplementary file 1E*, $\bar{x}$=3.6±4.4%, n=139 embryos, 3 experiments).

## Eye formation is compromised in *tcf7l1a* mutants when Tcf7l2 lacks the region encoded by exon5

In the *zf55* (*exl*) mutant allele of *tcf7l2*, the first intron of the gene is retained in the mRNA which knocks down expression of the protein generated by the *tcf7l2* transcripts with reading frames starting in exon 1 (*Muncan et al., 2007*). We found that homozygous *tcf7l2$^{zf55}$* mutants do not show reduced eye size at 30hpf (*Figure 3B,G*, *Supplementary file 1C*, n = 14, data from one of three experiments yielding similar results). To assess if *tcf7l2* repressor activity can functionally compensate for loss of *tcf7l1a*, we incrossed double heterozygous *tcf7l1a$^{+/m881}$/tcf7l2$^{+/zf55}$* fish. However, we did not observe an obvious eye-size phenotype in *Ztcf7l1a/tcf7l2* double homozygous mutants, with eye size in these mutants similar to *Ztcf7l1a$^{-/-}$* eyes (*Figure 3C,G*. *Supplementary file 1C*, three experiments). Neither did injection of *tcf7l2* ATG morpholino in *Ztcf7l1a$^{-/-}$* embryos lead to any enhancement of the eye phenotype (not shown). These results suggest that the lack of severe eye phenotypes in *tcf7l2* and *Ztcf7l1a/tcf7l2* double homozygous mutants is not due to genetic compensation, as has been observed for some other genes (*El-Brolosy et al., 2019*; *Ma et al., 2019*).

To further explore if the *45L-tcf7l2* splice variant could play a role in eye formation, we used 1.25 pmol/embryo of a splicing morpholino (mo$^{SPtcf7l2}$) that targets the intron/exon splice boundary 5' to exon 5 (*Figure 3—figure supplement 2A*). This Mo is predicted to force the splicing machinery to skip exon 5, such that only 4L-Tcf7l2 variants are translated. The efficacy of the mo$^{SPtcf7l2}$ was confirmed by RT-PCR, which shows that the cDNA fragment corresponding to *45L-tcf7l2* mRNA is absent in mo$^{SPtcf7l2}$ injected morphants, but the band corresponding to *4L-tcf7l2* mRNA is still present (*Figure 3—figure supplement 2B*). Sequencing of the putative *4L-tcf7l2* cDNA fragment showed that the exon4/6 splicing event in mo$^{SPtcf7l2}$ injected morphants is in frame and only leads to the expression of the *4L-tcf7l2* variant mRNA (not shown). As expected, Tcf7l2 protein was still present when detected by Western blot (*Figure 3—figure supplement 2B*).

Wildtype embryos injected with mo$^{SPtcf7l2}$ showed smaller eyes at 32hpf compared to control morpholino (mo$^{C}$) injected embryos (*Figure 3—figure supplement 2C-E*, n=10, *Supplementary file 1F*) and more dramatically, injection of mo$^{SPtcf7l2}$ in *Ztcf7l1a$^{-/-}$* mutants led to a fully penetrant eyeless phenotype ( *Figure 3F, H*, *Supplementary file 1D*, $\bar{x}$=99%, n=128, three experiments). No eyeless phenotype was observed when mo$^{SPtcf7l2}$ was injected in sibling heterozygous embryos or when a scrambled control morpholino (mo$^{C}$) was injected in *Ztcf7l1a$^{-/-}$* mutants (not shown). These results suggest that specific loss of exon5 coded sequence in Tcf7l2 can compromise its ability to promote eye formation. The eyeless phenotype in *Ztcf7l1a$^{-/-}$/mo$^{SPtcf7l2}$* is perhaps surprising given that *Ztcf7l1a$^{-/-}$/tcf7l2$^{zf55/zf55}$* embryos showed no severe eye phenotype, and suggests that an appropriate balance in the levels of *tcf7l2* splice variants may be required for the eye to form normally in *tcf7l1a* mutants.

## Tcf7l2 splice variant including exon five coding sequence shows repressor activity in luciferase reporter assays

Given the diversity of biological outputs driven by Wnt/β-catenin signalling and mediated by Tcf transcription factors (*Hoppler and Waterman, 2014*), the function of Tcf7l2 variants could potentially be to activate or repress the transcription of different subsets of genes. To explore whether 4L and 45L-Tcf7l2 variants show differing promoter transactivation abilities, we performed luciferase reporter assays using the generic TOPflash reporter and known promoters of the Wnt pathway regulated genes, *cdx1* (*Hecht and Stemmler, 2003*), *engrailed* (*McGrew et al., 1999*), *cJUN* (*Nateri et al., 2005*), *lef1* (*Hovanes et al., 2001*) and *siamois* (*Brannon et al., 1999*). All the

promoters of these genes used in the luciferase reporter assays contain consensus Tcf binding elements.

HEK293 cells were transiently transfected with the luciferase reporter construct and DNA encoding either: 1. the GSK-3β binding domain of mouse Axin2 Flag tag fusion (Flag-Ax2) construct which competes with GSK-3β leading to increased β-catenin levels and leads to Wnt pathway activation (*Figure 4A–F*; FlagAx-(501-560) in *Smalley et al., 1999*), or 2. a constitutively-active VP16-TCF7L2 fusion protein, which can induce the expression of Wnt-responsive promoters in absence of nuclear β-catenin (*Figure 4G–L*; *Ewan et al., 2010*). As expected, all the tested reporters showed a strong response to Flag-Ax2 or VP16-TCF7L2 transfection (*Figure 4A–L*, second bar in all plots, *Supplementary file 1G* and *Supplementary file 1H*). We then assessed whether Tcf7l2 splice variants with or without exon five could influence the luciferase reporter activation by co-transfecting *Flag-Ax2* (*Figure 4A–F*, *Supplementary file 1G*) or *VP16-TCF7L2* (*Figure 4G–L*, *Supplementary file 1H*) together with either *4L-tcf7l2* or *45L-tcf7l2* splice variants. 45L-Tcf7l2 variant expression led to reduced transactivation by FlagAx2 compared to 4L-Tcf7l2 on all the promotors we tested (*Figure 4A–F*, *Supplementary file 1G*). Moreover, 45L-Tcf7l2 also showed a greater ability to compete with VP16-TCF7L2, compared to 4L-Tcf7l2 when tested with all promotors except for *cjun*, which did not respond to any *tcf7l2* variant co-transfection (*Figure 4G–L*, *Supplementary file 1H*). Our results suggest that Tcf7l2 variants including exon5 are either less able to activate transcription or are able to repress the transcription at certain promoters.

## Inclusion of exon five encoded sequence enhances the interaction between Tcf7l2 and Tle3b

The region of Tcf7l2 encoded by exon five is located in the vicinity where Lef/Tcf proteins interact with Gro/TLE co-repressors (*Roose et al., 1998*; *Brantjes et al., 2001*; *Daniels and Weis, 2005*; *Arce et al., 2009*). Consequently, inclusion of exon five in *tcf7l2* mRNA could potentially modulate repressor activity by modifying the interaction of Tcf7l2 with Gro/TLE proteins. Alternatively, inclusion of exon five could modify the capacity of Tcf7l2 to interact with transactivating β-catenin.

To address if the inclusion of exon5-encoded sequence can modulate the interaction between Tcf7l2 and Gro/TLE or β-catenin proteins, we performed yeast two-hybrid (Y2H) protein interaction experiments between 4L-Tcf7l2 and 45L-Tcf7l2 variants and Tle3b (the zebrafish orthologue of mammalian Tle3/Gro1), or β-catenin (*Figure 5—figure supplement 1*). Full-length Tle3b seemed to be either toxic or have transfection problems in the yeast strain, and so we used a C-terminal deletion of Tle3b (dC-Tle3b), which still includes the glutamine-rich domain that interacts with Tcf proteins (*Daniels and Weis, 2005*). Yeast co-transfected with β-*catenin* or *dC-tle3b* and *4L-tcf7l2* or *45L-tcf7l2* splice variants were able to grow in complete auxotrophic selective media (-L-A-H-W + Aureoblastinin), and also express the X-gal selection reporter (*Figure 5—figure supplement 1*). This suggests that both Tcf7l2 variants are able to interact with β-catenin and dC-Tle3b. However, this Y2H assay cannot reveal differences in the affinity of interactions between proteins.

To address possible differences in protein interactions between Tcf7l2 splice variants and β-catenin or Tle3b, we performed co-immunoprecipitation (co-IP) experiments using protein extracts from co-transfected HEK293 cells. Both Myc-tagged Tcf7l2 variants were efficiently immunoprecipitated by anti-Myc beads (*Figure 5*, right top blot) and showed a similar capacity to co-IP with β-catenin (*Figure 5*, right middle blot, second and third lanes). However, MT-45L-Tcf7l2 showed a significantly higher capacity to co-IP Tle3b compared to 4L-Tcf7l2 (*Figure 5*, right bottom blot, second and third lane). This suggests that the repressor activity of MT-45L-Tcf7l2 observed in our in vivo and luciferase reporter assays could be mediated by an enhanced interaction with Gro/TLE co-repressors.

## Phosphorylated amino acids in the domain of Tcf7l2 encoded by exon five mediate transcriptional repressor function

Kinases are known to modulate Wnt signalling activity through phosphorylation of LRP6, β-catenin and Tcfs (*Li et al., 2002*; *Davidson et al., 2005*; *Zeng et al., 2005*; *Sokol, 2011*) and bioinformatic analysis using Netphos3.1 (cbs.dtu.dk/services/NetPhos; *Blom et al., 2004*) or GPS2.0 (gps.biocuckoo.org; *Xue et al., 2008*) predict that Tcf7l2 exon five coded Thr[172] and Ser[175] are putatively phosphorylated by GSK-3, CK1, PKA, and other kinases. GSK-3 and CK1 are known to regulate Wnt signalling by phosphorylating LRP6 co-receptor and β-catenin (*Niehrs, 2012*).

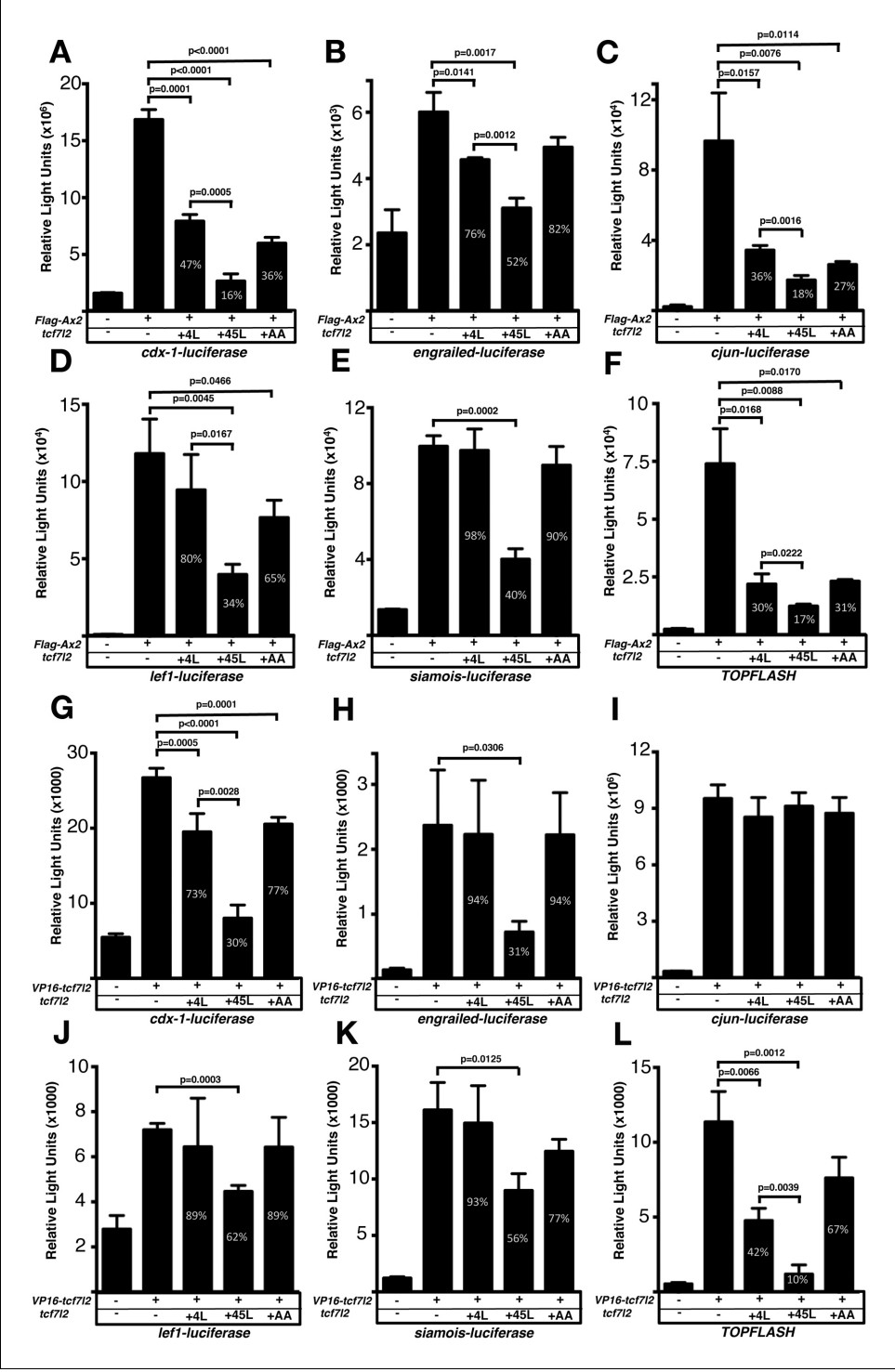

**Figure 4.** Exon five coding *tcf7l2* variant represses Wnt activity induced by *FLAG-Ax2* or *VP16-TCF7L2* in luciferase assays. Bar plots showing luciferase reporter assay results expressed in relative light units. HEK293 cells were transiently co-transfected with luciferase reporter constructs indicated beneath the X-axis, (A–F) *FLAG-Ax2* (except for first bars), (G–L) *VP16-TCF7L2* DNA (except for first bars) and *4L-tcf7l2* DNA (+4L; 3rd bars), or *45L-tcf7l2* DNA (+45L; 4th bars) or *tcf7l2-AA* DNA (+AA; 5th bars). Control experiments show only background luciferase activity with no transfected plasmids (1st bars). Figures in the bars indicate the percentage size of that bar relative to transfection with either *FLAG-Ax2* or *VP16-TCF7L2* alone (2nd bar). Error bars are mean ± SD n = 3 (experiments
*Figure 4 continued on next page*

*Figure 4 continued*

were performed twice), P values from unpaired t tests comparing *FLAG-Ax2* or *VP16- TCF7L2* control condition with *tcf7l2* variant co-transfections. Comparisons with no statistical significance are not marked.

To address if amino acids in the Tcf7l2 exon five encoded region are phosphorylated, we performed mass spectrometry (MS) analysis. N-terminal *MT-4L-tcf7l2* and *MT-45L-tcf7l2* variants were cloned in frame to BioID2 (*Kim et al., 2016*), an optimised biotin ligase protein, and were transiently expressed in HEK cells treated with biotin. Steptavidin pulled-down proteins were used for liquid chromatography with tandem mass spectrometry analysis recovering 40% of Tcf7l2 amino acid sequence (*Figure 5—source data 1*). The recovered peptide containing exon 5-coded sequence, Asp$^{151}$-Lys$^{184}$ was phosphorylated (*Supplementary file 1I*).

To address the role of the putatively phosphorylated exon 5 coded amino acids in Tcf7l2, we generated an MT-45L-Tcf7l2 mutant version in which both Thr$^{172}$ and Ser$^{175}$ were replaced by alanines (MT-45L-Tcf7l2-AA), which cannot be phosphorylated. Unlike MT-*45L-tcf7l2*, expression of *MT-45L-tcf7l2-AA* (20pg mRNA per embryo) was unable to rescue the eyeless phenotype of *Ztcf7l1a$^{-/-}$/ tcf7l1b* morphant embryos (*Figure 3H*, *Supplementary file 1E*, $\bar{x}$=12.4±2.5%, n=117, three experiments). However, *MT-45L-tcf7l2* mutant forms containing the mutant variants Thr172Ala (20pg MT-

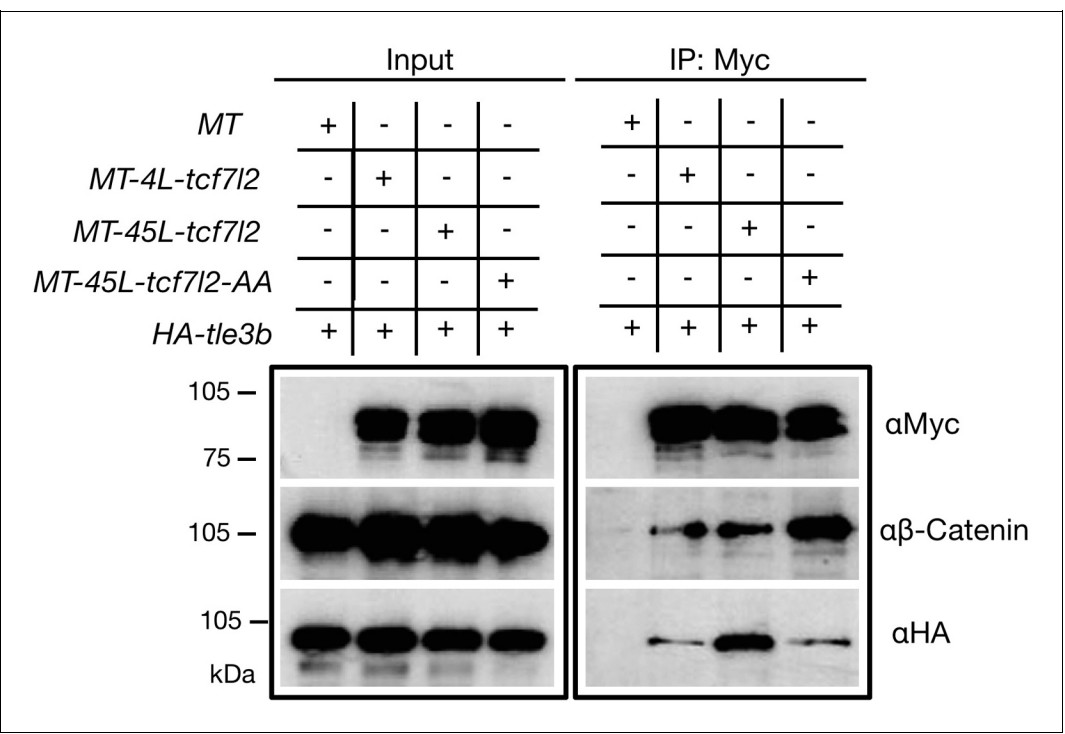

**Figure 5.** Alternative exon 5 of *tcf7l2* enhances affinity with Tle3b. Protein input (left panel) and anti-Myc immunoprecipitation (IP) eluate western blot (right panel) showing co-inmunoprecipitation of β-Catenin or HA-tagged Tle3b. HEK293 cells were transiently transfected with HA tagged *tle3b* together with empty myc tag vector (1$^{st}$ lane), *MT-4L-tcf7l2* (2$^{nd}$ lane), *MT-45-tcf7l2* (3$^{rd}$ lane) and *MT-AA-tcf7l2* (4$^{th}$ lane). Left panels show protein input before anti-Myc IP. Right panels show protein eluate from anti-Myc antibody coupled beads. Westernblots were probed with anti-Myc (tagged Tcf7l2 proteins, top panel), anti-βcatenin (middle panel) and anti-HA (tagged Tle3b protein, bottom panel) antibodies. Asterisk shows that the Tcf7l2 form containing exon five shows more intense binding with Tle3b than other Tcf7l2 forms.

The online version of this article includes the following source data and figure supplement(s) for figure 5:

**Source data 1.** Zebrafish MT-45L-Tcf7l2-BioID2 peptides recovered by LC-MS/MS analyses are shown in bold.

**Figure supplement 1.** 4L-Tcf7l2 and 45L-Tcf7l2 variants interact with β-Catenin and Tle3b in yeast two-hybrid protein interaction assays.

45L-tcf7l2-AS mRNA per embryo, *Figure 3H*, *Supplementary file 1E*, $\bar{x}$=76.6±0.5%, n=34, two experiments) or Ser175Ala (20pg MT-*45L-tcf7l2-TA* mRNA per embryo, *Figure 3H*, *Supplementary file 1E*, $\bar{x}$=77.9±17.9%, n=44, two experiments), were able to rescue the eyeless phenotype suggesting that repressor activity could be elicited through phosphorylation of either or both of amino acids Thr[172] and Ser[175]. Furthermore, we generated a Thr172Glu/Ser175Glu phospho-mimicking MT-45L-Tcf7l1a mutant form of 45L-Tcf7l2 (MT-45L-Tcf7l2-EE). Expression of *MT-45L-tcf7l2-EE* (20pg mRNA per embryo) restored eye formation in 80.3±4.5% of *Ztcf7l1a*[-/-]*/tcf7l1b* morphant embryos (*Figure 3H*, *Supplementary file 1E*, n=48, three experiments).

Moreover, MT-45L-Tcf7l2-AA behaved like MT-4L-Tcf7l2 variants in luciferase reporter assays (*Figure 4*, fifth bar in all plots, *Supplementary file 1 G and H*), and also showed a reduced capacity to interact with Tle3b in co-IP experiments (*Figure 5*, right bottom blot, fourth lane). Of note, the MT-Tcf7l2-AA mutant variant showed a greater capacity to co-IP with β-catenin compared to MT-4L-Tcf7l2 or MT-45L-Tcf7l2 variants (*Figure 5*, right middle blot, fourth lane). These results suggest that phospho Thr[172] and phospho Ser[175] residues may regulate the ability of 45L-Tcf7l2 to repress target genes and consequently contribute, together with Tcf7l1a/Tcf7l1b function, to eye specification in zebrafish.

## Discussion

In this study, we show that developmentally regulated splicing contributes to Wnt signalling regulation during patterning of the anterior neural plate and eyes. We show that there is extensive variety in splicing of zebrafish *tcf7l2* throughout development and across adult tissues and that the exon five encoded region within Tcf7l2 influences its transcriptional repressor activity. Tcf repressor function is required for maintaining low levels of pathway activity during forebrain and eye specification. We show that long carboxy-terminal Tcf7l2 variants that include both exons 4 and 5 can restore eye formation in embryos with compromised *tcf7l1a/b* function and that specific knockdown of the 45L-Tcf7l2 variant in *tcf7l1a* mutants leads to embryos with no eyes. A similar specific human Tcf7l2 variant that includes the region encoded by the alternative exon 3a is also able to restore eye formation in *tcf7l1a/b* knockdown embryos, suggesting conservation in the role of splicing in mediating transcriptional repressor activity of Tcf7l2 proteins. Additionally, 45L-Tcf7l2 variants have less capacity to transactivate known Wnt target gene promoters in luciferase reporter assays and can out-compete constitutively transcriptionally active *VP16-hTcf7l2* chimeric proteins. Liquid chromatography with tandem mass spectrometry results also suggests that the protein region coded by *tcf7l2* exon five is phosphorylated and that the transcriptional repressor function of the 45L-Tcf7l2 variant requires this phosphorylation potentially for a more efficient interaction with Tle3b corepressor.

### Tcf7l2 transcriptional repression function contributes to forebrain patterning

Like *tcf7l1a* and *tcf7l1b*, *tcf7l2* is expressed in the anterior neural ectoderm, from where the eyes and forebrain develop (*Kim et al., 2000*; *Young et al., 2002*; *Dorsky et al., 2003*). However, their expression patterns are not completely identical as *tcf7l2* expression does not include the area of strong expression of *tcf7l1a* and *tcf7l1a* in the presumptive telencephalon or the midbrain-hindbrain boundary (*Dorsky et al., 2003*). These differences could contribute to the non-redundant functions between Lef/Tcf transcription factors. Experimental manipulations or mutations that increase Wnt/β-catenin activity in the anterior neural plate during gastrulation generate embryos with no forebrain and eyes (*Wilson and Houart, 2004*). For instance, *tcf7l1a* is cell-autonomously required for eye field specification and zebrafish embryos lacking zygotic *tcf7l1a/headless*[m881] function have smaller eyes (*Kim et al., 2000*; *Young et al., 2019*). However, this phenotype is exacerbated when *tcf7l1b* is knocked down in *tcf7l1a* mutants leading to embryos with no eyes (*Dorsky et al., 2003*). The notion that active repression by Tcf7l1a/b transcription factors is required for neuroectodermal patterning is supported by the observation that overexpression of transcriptional dominant active *VP16-tcf7l1a* chimera also leads to eyeless embryos (*Kim et al., 2000*).

Eyes are smaller in wild-type embryos and absent in *Ztcf7l1a*[-/-] embryos in which *tcf7l2* variants including exon5 have been knocked down. As only the 45L-Tcf7l2 variant, which includes exon5, can efficiently restore eye formation in *Ztcf7l1a*[-/-]*/tcf7l1b* morphant embryos, our results suggest that the

exon five coding region can assign transcriptional repressor activity to Tcf7l2 and, as for Tcf7l1 proteins, this transcriptional repressive function contributes to forebrain patterning and eye formation. This conclusion is supported by 45L-Tcf7l2 variants having less transactivation capacity and greater repressor activity compared to 4L-Tcf7l2 and 45L-Tcf7l2-AA variants in luciferase reporter experiments; additionally, 45L-Tcf7l2 variants show greater association for Tle3b transcriptional co-repressor in co-IP experiments. Hence, the inclusion of exon5 in *tcf7l2* variants could potentially influence the balance between activation *versus* repression transcription activity of Tcf7l2.

The lack of severe forebrain and eye phenotypes in single *tcf7l1* mutants is at least in part due to the overlapping functional activities of different *tcf* genes and we assume the same is likely for the *tcf7l2* mutant. However, we find no evidence that mutant mRNA triggered genetic compensation (*El-Brolosy et al., 2019*; *Ma et al., 2019*; *El-Brolosy and Stainier, 2017*; *Rossi et al., 2015*) is the reason that *tcf7l2* mutants lack severe phenotypes. For instance, morpholino knockdown of *tcf7l2* translation, which circumvents this mechanism, does not result in a severe eye phenotype. Given that, as for other Tcf genes (*Cadigan, 2012a*; *Ramakrishnan et al., 2018*), *tcf7l2* specific splice variants seem to encode proteins with either repressor or activator roles, mutations that disrupt the balance between these functions may lead to phenotypes differing from complete gene abrogation. Potentially, misregulation of repressor function while maintaining the ability of the protein to activate transcription (as suggested by our experiments that specifically knockdown *45L-tcf7l2* variants with a splicing morpholino) might lead to forebrain respecification and small or no eye phenotypes that differ from complete loss of protein expression. Hence, the balance between the levels of Tcf transcriptional repressing *versus* activating variants may facilitate the correct subdivision of the telencephalic, eye and diencephalic territories of the neural plate.

## Phosphorylation contributes to the balance of transcriptional activation-repression by mediating the function of Tcf7l2 as a transcriptional repressor

The context-dependent regulatory domain of Tcfs, which includes the region coded by *tcf7l2* exon 5, can be acetylated, phosphorylated or sumoylated (*Yamamoto et al., 2003*; *Shetty et al., 2005*; *Mahmoudi et al., 2009*; *Hikasa et al., 2010*; *Ota et al., 2012*; *Elfert et al., 2013*). The importance of phosphorylation to the repressor activity of 45L-Tcf7l2 is suggested by the finding that the phosphorylation resistant 45-L-Tcf7l2-AA mutant form, in which both $Thr^{172}$ and $Ser^{175}$ amino acids are replaced by alanines, behaves as if lacking the region coded by exon five in all the functional assays we tested. Conversely, the phospho-mimicking EE-Tcf7l2 variant was able to restore eye development in $Ztcf7l1a^{-/-}/tcf7l1b$ morphant embryos suggesting that phosphorylation of either $Thr^{172}$ or $Ser^{175}$ is required for the repressive function of 45L-Tcf7l2 variants. Although our mass spectrometry analysis confirmed that the peptide encoded by zebrafish *tcf7l2* exon five is phosphorylated, the resolution was not sufficient to discriminate whether the phosphorylation was on $Thr^{172}$, $Ser^{175}$ or both. However, mRNA overexpression of 45L-Tcf7l2-AS or 45L-Tcf7l2-TA mutant variants can restore eye formation in $Ztcf7l1a^{-/-}/tcf7l1b$ morphant embryos, suggesting that the phosphorylation of either amino acid is sufficient to enable the repressor activity of 45L-Tcf7l2.

Gro/TLE transcription co-repressors are displaced by β-catenin to allow Tcf-mediated transcriptional activation (*Daniels and Weis, 2005*; *Arce et al., 2009*; *Chodaparambil et al., 2014*), and although we find 45L-Tcf7l2 still interacts with β-catenin, the 45L-Tcf7l2-AA mutant variants show greater binding capacity. This raises the possibility that 45L-Tcf7l2 variants may co-exist as phosphorylated and un-phosphorylated pools with different repressor/activator activity. This may also help explain why overexpression of moderate levels of 45L-Tcf7l2 shows no overt effect in wildtype embryos, possibly because Tcf7l2 phosphorylation and not its protein levels alone may control the activation-repression balance of these variants and suggesting that the phosphorylation of 45L-Tcf7l2 is a permissive event.

It is widely accepted that Tcfs work as transcriptional switches, repressing transcription of downstream genes in absence of Wnt ligand, and activating gene transcription when Wnt signalling is active (*Cadigan, 2012a*; *Ramakrishnan et al., 2018*). In this context, the repressive function of Tcf7l2 may be part of an integrated pathway response when the Wnt pathway is not active. The 'off' state of Wnt signalling involves active phosphorylation of β-catenin by CK1α and GSK-3β kinases (*MacDonald and He, 2012*; *Niehrs, 2012*). Our findings support a model in which Tcf transcription factors could also be part of a kinase regulatory module that maintains the pathway in an 'off' state,

not only in the cytoplasm by phosphorylating β-catenin, but also by promoting transcriptional repression through phosphorylation of Tcf7l2. Direct assessment of an in vivo functional role for phosphorylation of Tcf7l2, and potentially other Tcfs, will be required to address such a model.

The salience of resolving a role for phosphorylation in the regulation of transcriptional activation/repression is heightened given the relevance of Tcf7l2 in mediating colorectal cancer outcome due to imbalanced Wnt signalling. Indeed, one future avenue for investigation will be to identify proteins that interact with the phosphorylated and un-phosphorylated forms of Tcf7l2.

## Functional modulation of Tcf7l2 through spatial and temporal regulation of splicing of alternative exons

The occurrence of widespread tissue-specific and developmentally regulated *tcf7l2* splicing suggests that certain variants of Tcf7l2 are required for proper cell and tissue type specification during development, and for its various roles in organ physiology during adult life (*Nusse and Clevers, 2017*). For instance, we have previously shown that Tcf7l2 is required for the development of left-right asymmetry of habenular neurons (*Hüsken et al., 2014*) and at the stages during which habenular neurons are becoming lateralised, *tcf7l2* splice variants transition from expressing long to only medium and short carboxy-terminal end variants. This is potentially significant as only long variants include a complete C-clamp DNA binding-helper domain and the C-terminal binding protein domains (*Young et al., 2002*). The C-clamp domain can direct Tcf7l2 to specific promoters (*Hoverter et al., 2014*). Consequently, absence of a whole C-clamp may bias the promoter occupancy of Tcf7l2 and shift the expression profile of Wnt target genes (*Atcha et al., 2003*; *Atcha et al., 2007*; *Hecht and Stemmler, 2003*; *Hoverter et al., 2014*).

Tcf7l2 is linked to type-2 diabetes outcome (*Grant et al., 2006*; *Lyssenko et al., 2007*; *Prokunina-Olsson et al., 2009*; *Savic et al., 2011*) and the strongest risk factor SNPs are located in introns near human *tcf7l2* exon 3a (*Grant et al., 2006*). Perhaps surprisingly, only liver tissue-specific *tcf7l2* knockouts in mice and no other organs (including pancreas), lead to metabolic outcomes mimicking type-2 diabetes (*Boj et al., 2012*; and see *Bailey et al., 2015*). Furthermore, a recent study in rats has shown that Tcf7l2 in habenular neurons regulates nicotinic acetylcholine receptor function, concomitantly impacting pancreatic function and glucose homeostasis (*Duncan et al., 2019*). At least in zebrafish, the brain appears to express predominantly medium and short C-terminal *Tcf7l2* splice variants in combination with all kinds of splice variants in the region of alternative exons 4 and 5. It will be interesting to resolve the specific expression of *tcf7l2* splice variants in the habenula and assess whether these have function in the regulation of nicotinic receptor function.

Although homologies between fish and human exons in the region of Tcf7l2 alternative exons 4/5 are uncertain and despite lack of overall sequence conservation, human *tcf7l2* exon 3a and fish exon five have a similar size and share amino acids that are likely phosphorylated in zebrafish Tcf7l2 encoded exon five region. Moreover, human *tcf7l2* variants including alternative exon 3a, and to a lesser extent exon 4, can restore eye formation in *Ztcf7l1a⁻/⁻/tcf7l1b* knockdown embryos. This suggests that the region encoded by human Tcf7l2 exon 3a, and possibly exon 4a, may have similar functions as zebrafish exon five and that alternative exons in this region may play an evolutionarily conserved role in the transcriptionally repressive capacity of Tcf7l2. Consequently, it will be interesting to assess if the expression levels of these human *TCF7L2* exons are altered in type-2 diabetes patients carrying risk factor SNPs.

Given the importance of maintaining balanced Wnt/β-catenin pathway activity throughout development and tissue homeostasis, elucidating all mechanisms that impact Wnt signalling modulation is critical if we are to understand and develop ways to manipulate pathway activity when misregulated in pathological conditions. Our work adds weight to the idea that the regulation of alternative splicing and controlling the balance between repressor and activator functions of Tcf proteins play an important role in Wnt/β-catenin pathway regulation.

## Materials and methods

### Animal use, mutant and transgene alleles, genotyping and quantification of eye size

Adult zebrafish were kept under standard husbandry conditions and embryos obtained by natural spawning. Wildtype and mutant embryos were raised at 28°C and staged according to *Kimmel et al. (1995)*. Fish lines used were *tcf7l1a^m881* (*Kim et al., 2000*), *tcf7l1b^zf157tg* (*Gribble et al., 2009*) and *tcf7l2^zf55* (*Muncan et al., 2007*). These three lines are likely to abrogate expression of proteins coded by the reading frame starting in exon 1. There is an alternative downstream transcription start site in mouse *tcf7l2* (*Vacik et al., 2011*) and likely in other *tcf* genes too (unpublished observations). It is not known if transcripts from these alternative start sites have any functional roles in embryos carrying the mutations above.

Genomic DNA was isolated by HotSHOT method (Suppl. Materials and methods) and *tcf7l1a^m881* and *tcf7l2^zf55* mutations were genotyped by KASP assays (K Biosciences, assay barcode 1145062619) using 1 μl of genomic DNA for 8 μl of reaction volume PCR as described by K Biosciences. Adult zebrafish organs were dissected as described in Supp. Materials and methods.

The sizes of eye profiles were quantified from lateral view images of PFA fixed embryos by delineating the eye using Adobe Photoshop CS5 magic wand tool and measuring the area of pixels included in the delineated region. The surface area was then transformed from $px^2$ to $\mu m^2$ and then to predicted eye volume as in *Young et al. (2019)*. Embryos were scored as eyeless when no retinal tissue was observed. Even though in the rescue experiments there was variability in the size of the restored eyes, for the sake of simplicity, all embryos with distinguishable eyes were scored as having eyes. This binary categorisation made the rescued versus not-rescued eye phenotype more straightforward to score and less affected by subjective interpretation.

### RNA extraction, reverse transcription and PCR

Total RNA was extracted from live embryos and adult zebrafish using Trizol (Invitrogen) and homogenised by pestle crushing and vortexing. SuperscriptII (Invitrogen) was used for reverse transcription under manufacturers' instructions using oligo dT and 1 μg of RNA for 20 μg reaction volume. The following primers were used to amplify fragments of *tcf7l2* cDNA: region exon4/5 (Set a-F TCAAAACAGCTCTTCGGATTCCGAG, Set a-R CTGTAGGTGATCAGAGGTGTGAG), region exon15 (Set b-F GATCTGAGCGCCCCAAAGA AGTG Set b-R CGGGGAGGGAGAAATCATGGA GG).

### mRNA synthesis, embryo microinjection and morpholinos

*tcf7l2* splice variant PCR fragments were cloned in pCS2+ or pCS2+MT expression vectors for mRNA synthesis. mRNA for overexpression was synthesised using SP6 RNA mMessage mMachine transcription kit (Ambion). One to two cell stage embryos were co-injected with 10 nl of 5 pg of GFP mRNA and morpholinos or in vitro synthesised mRNA at the indicated concentrations. Only embryos with an even distribution of GFP fluorescence were used for experiments. Morpholino sequences: mo^tcf7l2ATG (5'-CATTTTTCCCGAGGAGCGCTAATTT-3'). Embryos injected with this morpholino fail to produce Tcf7l2 protein (*Figure 3—figure supplement 1B*; ).

mo^SPtcf7l2 (5'-GCCCCTGCAAGGCAAAGACGGACGT-3'). This splice-blocking morpholino leads to exon skipping to generate a Tcf7l2 protein lacking exon five derived amino acids (Figure S5). *tcf7l1a^-/-* embryos injected with mo^SPtcf7l2 lack eyes, a phenotype not seen when the morpholino is injected into *tcf7l1a^+/-* siblings. No equivalent genetic mutation that leads to loss of exon 5 of *tcf7l2* exists but the loss of eye phenotype is consistent with other conditions in which the overall level of TCF-mediated repression is reduced (*Dorsky et al., 2003*).

moC (5'- CTGAACAGGCGGCAGGCGATCCACA −3'). This morpholino is a sequence scrambled version of mo^SPtcf7l2 used as an injection control.

mo^tcf7l1b (5'-CATGTTTAACGTTACGGGCTTGTCT-3'; *Dorsky et al., 2003*). *tcf7l1a^m881/m881* embryos injected with mo^tcf7l1b phenocopy the loss of eye phenotype seen in *tcf7l1a^m881/m881*/ *tcf7l1b^zf157tg/zf157tg* double mutants (Young and Wilson, unpublished).

## In situ hybridisation and probe synthesis

Digoxigenin (DIG) and fluorescein (FLU)-labelled RNA probes were synthesized using T7 or T3 RNA polymerases (Promega) according to manufacturers' instructions and supplied with DIG or FLU labelled UTP (Roche). Probes were detected with anti-DIG-AP (1:5000, Roche) or anti-FLU-AP (1:10000, Roche) antibodies and NBT/BCIP (Roche) or INT/BCIP (Roche) substrates according to standard protocols (*Thisse and Thisse, 2008*).

## Luciferase reporter experiments

The following reporters were used: *cdx1*:luc (*Hecht and Stemmler, 2003*), *engrailed*:luc (*McGrew et al., 1999*), *cJun*:luc (*Nateri et al., 2005*), *lef1*:luc (*Hovanes et al., 2001*), *siamois*:luc (*Brannon et al., 1999*), and TOPflash (*Molenaar et al., 1996*). HEK cells were transfected according to standard methods and using the conditions described in Supp. Materials and methods.

## Zebrafish protein extraction

Embryos were washed once with chilled Ringers solution, de-yolked by passing through a narrow Pasteur pipette, washed three times in chilled Ringers solution supplemented with PMF (300 mM) and EDTA (0.1 mM). Samples were briefly spun down, media removed, Laemlli buffer 1X was added at 10 μl per embryo and incubated for 10 min at 100°C with occasional vigorous vortexing before chilling on ice. Samples were loaded in polyacrylamide gels or stored at −20°C.

## HEK293 cell line transfection, immunohistochemistry and co-immunoprecipitation

Authenticated HEK293 cells were purchased from ATCC and tested for mycoplasma by PCR. HEK293 cells were grown in six well plates and transfected with 4 μg of each DNA with lipofectamine 2000 (Invitrogen) for 6 hr according to manufacturers instructions.

For immunohistochemistry, cells were fixed 48 hr after transfection in 4% paraformaldehyde in PBS for 20 min at room temperature, washed with PBS, and permeabilized in 0.2% Triton X-100 in PBS for 5 min at room temperature. The protocol was followed as in Supp. Materials and methods.

For immunoprecipitation, cells were grown for 24 hr after transfection and then proteins were extracted following standard methods (Supp. Materials and methods). The eluate from antibody beads (30 μl) was loaded in 10% polyacrylamide gels and proteins were detected by Western blots (standard conditions) using anti-myc (1/20,000, SC-40, SCBT), anti-HA (1/10,000, 3F10, Roche) and anti-β-catenin (1/8000, Sigma, C7207), to detect the co-immunoprecipitated proteins. Antibodies used on Western blots in *Figure 3—figure supplement 1B* are anti human Tcf7l2 (N-20, SCBT) and anti-gamma tubulin (T9026, Sigma) HRP coupled secondary antibodies (1/2,000, sigma) were used and blots were developed using an ECL kit (Promega).

## Mass Spectrometry experiments

Cell extract proteins were pulled down with streptavidin coated magnetic beads. The protein eluate was run on an SDS-PAGE gel and stained with Coomassie blue. Stained gels were cut, de-stained and Trypsin Gold, Mass Spectrometry Grade (Promega, Madison, USA) in 50 mM ammonium bicarbonate was added in each well containing dried gel pieces and incubated overnight at 37°C. Next day, 0.1% formic acid was added to stop the trypsinolysis and the eluted tryptic peptides were collected in MS glass vials, vacuum dried and dissolved in 0.1% formic acid for LC-MS/MS.

LC-MS/MS analysis was performed with an LTQ-Velos mass spectrometer (Thermo Fisher Scientific, U.K.). Peptide samples were loaded using a Nanoacquity UPLC (Waters, U.K.) with Symmetry C18 180umX20mm (Waters part number 186006527) trapping column for desalting and then introduced into the MS via a fused silica capillary column (100 μm i.d.; 360 μm o.d.; 15 cm length; 5 μm C18 particles, Nikkyo Technos CO, Tokyo, Japan) and a nanoelectrospray ion source at a flow rate at 0.42 μl/min. The mobile phase comprised $H_2O$ with 0.1% formic acid (Buffer A) and 100% acetonitrile with 0.1% formic acid (Buffer B). The gradient ranged from 1% to 30% buffer B in 95 min followed by 30% to 60% B in 15 min and a step gradient to 80% B for 5 min with a flow of 0.42 μl/min. The full scan precursor MS spectra (400–1600 m/z) were acquired in the Velos-Orbitrap analyzer with a resolution of r = 60,000. This was followed by data-dependent MS/MS fragmentation in centroid mode of the most intense ion from the survey scan using collision induced dissociation (CID) in the

linear ion trap: normalized collision energy 35%, activation Q 0.25; electrospray voltage 1.4 kV; capillary temperature 200°C: isolation width 2.00. The targeted ions were dynamically excluded for 30 s and this MS/MS scan event was repeated for the top 20 peaks in the MS survey scan. Singly charged ions were excluded from the MS/MS analysis and XCalibur software version 2.0.7 (Thermo Fisher Scientific, U.K.) was used for data acquisition. Raw data were analysed using Proteome Discoverer (PD v1.3) with Mascot search engine and Swiss-Prot human and Zebrafish proteome database. Up to two trypsin missed cleavages were allowed, carbamidomethylation was set as a fixed modification, while methionine oxidation, phosphorylation of serine, threonine and tyrosine were set as variable modifications. Mass tolerance was set to eight ppm for the precursors and to 0.6 Da for the fragments.

### Yeast two-hybrid assays

N-terminal deletions of the first 53 amino acids of *tcf7l2* splice variants were cloned in *pGBK*. Full-length β-catenin and a C-terminal deletion of *tle3b* (NM_131780, complete reading frame after amino acid 210) were cloned in *pGAD* (Clontech). Combinations of plasmids to test two-hybrid interactions were co-transformed in Y2Gold yeast strain (Suppl. Materials and methods). Transformed yeast were plated on -Leu-Trp dropout selective media agar plates supplemented with X-gal. Positive blue colonies were streaked to an -Ade-His-Leu-Trp dropout selective media agar plates supplemented with Aureoblastidin A and X-gal (Clontech yeast two-hybrid manual).

## Acknowledgements

We thank members of our lab, Florencia Cavodeassi and Mate Varga for stimulating discussions, the UCL fish facility team for fish care and Elke Ober, Richard Dorsky, Marian Waterman, Randy Moon and others for reagents; Masa Kai for advice on Western blot methods and Abdol Nateri for advice on HEK cell protein extraction. We also thank Lucia di Vagno, Graham W Taylor and Mark Crawford for mass spectrometry analysis. This study was generously supported by the MRC (MR/L003775/1 to SW and Gaia Gestri) and Wellcome Trust (088175, 104682/Z/14/Z SW and RY), a Marie Curie Incoming International Fellowship (RY), a Royal Society International Joint Project (SW and MA) and FONDAP (15090007) and FONDECYT (1180606) to MA.

## Additional information

### Funding

| Funder | Grant reference number | Author |
| --- | --- | --- |
| Wellcome | 088175 | Rodrigo M Young<br>Stephen W Wilson |
| Royal Society | International Joint Project | Miguel L Allende<br>Stephen W Wilson |
| Medical Research Council | MR/L003775/1 | Stephen W Wilson |
| Wellcome | 104682/Z/14/Z | Stephen W Wilson |
| H2020 Marie Skłodowska-Curie Actions | Marie Curie Incoming International Fellowship | Rodrigo M Young |
| National Fund for Scientific and Technological Development | 1180606 | Miguel L Allende |
| Fund for Research Centres in Priority Areas | | Miguel L Allende |
| Cancer Research UK | C1295/A15937H | Kenneth B Ewan<br>Trevor C Dale |

The funders had no role in study design, data collection and interpretation, or the decision to submit the work for publication.

## Author contributions

Rodrigo M Young, Conceptualization, Data curation, Formal analysis, Investigation, Methodology, Writing—original draft, Writing—review and editing; Kenneth B Ewan, Veronica P Ferrer, Data curation, Formal analysis, Investigation; Miguel L Allende, Funding acquisition, Investigation; Jasminka Godovac-Zimmermann, Investigation, Writing—review and editing; Trevor C Dale, Supervision, Funding acquisition, Investigation; Stephen W Wilson, Supervision, Funding acquisition, Writing—original draft, Project administration, Writing—review and editing

## Author ORCIDs

Rodrigo M Young (ID) https://orcid.org/0000-0001-5765-197X
Kenneth B Ewan (ID) https://orcid.org/0000-0001-6622-9009
Veronica P Ferrer (ID) https://orcid.org/0000-0002-7687-6151
Miguel L Allende (ID) http://orcid.org/0000-0002-2783-2152
Jasminka Godovac-Zimmermann (ID) http://orcid.org/0000-0002-6820-4128
Trevor C Dale (ID) https://orcid.org/0000-0002-4880-9963
Stephen W Wilson (ID) https://orcid.org/0000-0002-8557-5940

## Ethics

Animal experimentation: Ethical approval for zebrafish experiments was obtained from the Home Office UK according to the Animal Scientific Procedures Act 1986.

## Decision letter and Author response

Decision letter https://doi.org/10.7554/eLife.51447.sa1
Author response https://doi.org/10.7554/eLife.51447.sa2

## Additional files

### Supplementary files

• Supplementary file 1. *Supplementary file 1A* Analysis of RT-PCR data in *Figure 1E* and Figure 2B, C showing the expression of alternative *tcf7l2* exons 4, 5 and 15 (a) and Tcf7l2 variants (b) through development. (a) Results in rows 1 to 4 (grey) taken from *Figure 1E*, rows 5 to 7 (pink) from *Figure 2B* and rows 8 to 10 (blue) from *Figure 2C*. (+) and (++) depict relative band intensity in the gels; (-) indicates no signal (b) Tcf7l2 variants expressed during development based on the information in (a). (+) and (-) indicate presence or absence of the variant respectively. (?) indicates that it is not possible to derive a conclusion based on the data available. Analysis of variants that lack both exons 4 and 5 is not included. *Supplementary file 1B* Analysis of RT-PCR data from Figure 2D-E showing the expression of alternative *tcf7l2* exons 4, 5 and 15 (a) and Tcf7l2 variants (b) in adult organs. (a) Results in rows 1 to 4 (grey) taken from *Figure 2D-E* rows 5 to 7 (pink) from *Figure 2F* and rows 8 to 10 (blue) from *Figure 2G*. (+) and (++) depict relative band intensity in the gels; (-) indicates no signal (b) Tcf7l2 variants expressed in adult organs based on the information in (A). (+) and (-) indicate presence or absence of the variant respectively. (?) indicates that it is not possible to derive a conclusion based on the data. Analysis of variants that lack both exons 4 and 5 is not included. *Supplementary file 1C* Size of the *tcf7l2$^{-/-}$*eye is similar to wildtype embryos at 30hpf. Estimated volume in $\mu m^3$ of the eye of 30hpf fixed embryos coming from a double heterozygous *tcf7l1a/tcf7l2* mutant incross. Avg, average; SD, Standard Deviation. *Supplementary file 1D* Knockdown of *tcf7l1b* and excision of *tcf7l2* exon5 in *tcf7l1a$^{-/-}$* mutant embryo compromises eye formation Embryos from female *tcf7l1a$^{+/-}$* to male *tcf7l1a$^{-/-}$* spawnings were injected with the morpholinos stated in the left column. This pairing scheme leads to 50% of homozygous mutant embryos. Each row represents an individual experiment. Embryos were scored as eyeless when little or no pigmented retinal tissue could be distinguished. Total represents the number of embryos scored in each experiment. *Supplementary file 1E* Restoration of eye formation by expression of exogenous Tcf7l2 variants in *tcf7l1a$^{-/-}$/tcf7l1b* morphant embryos. *Tcf7l1a$^{-/-}$* embryos injected with *tcf7l1b* morpholino and *tcf7l1a* or the *tcf7l2* mRNA variant stated in the first column. Each row represents an individual experiment. Total represents the number of *tcf7l1a$^{-/-}$* embryos scored in each experiment.

Eye formation was scored as rescued when pigmented retinal tissue was evident. *Supplementary file 1F* Size of the eye profile area is smaller in $mo^{SPtcf7l2}$ injected embryos at 30hpf. Volume in µm³ of the eye profile of 32hpf fixed embryos from wildtype embryos injected with $mo^C$ or $mo^{SPtcf7l2}$. Avg, Average; SD, Standard Deviation. *Supplementary file 1G* Results from luciferase reporter assay experiments expressed in relative light units using *FLAG-Ax2* to induce Wnt activity. Avg, Average; SD, Standard Deviation; %, percentage relative to *FLAG-Ax2* condition. *Supplementary file 1H* Results from luciferase reporter assay experiments expressed in relative light units using *VP16-TCF7L2* to induce Wnt activity. Avg, Average; SD, Standard Deviation; %, percentage relative to *VP16-TCF7L2* condition. *Supplementary file 1I* Peptides recovered by mass spectrometry and their respective modifications.

• Transparent reporting form

### Data availability

All data generated or analysed during this study are included in the manuscript and supporting files.

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

## Appendix 1

# Supplementary materials and methods

### Genotyping and adult tissue dissection

Genomic DNA was extracted from methanol or 4% paraformaldehyde fixed embryos by incubating in 25 µl of KOH 1.25M, EDTA 10 mM at 95°C for 30 min, and then neutralised with 25 µl of Tris-HCl 2M.

For tissue dissection, adult fish around a year old were culled by deep anesthetising in ice-cold 0.15% 2-phenotyethanol. Liver and pancreas tissue were isolated under a dissecting fluorescence microscope using Tg(fabp10:dsRed)$^{gz4}$ (**Dong et al., 2007**) and Tg(XlEef1a1: GFP)$^{s854}$ (**Field et al., 2003**) lines to distinguish liver and pancreas respectively. Other organs had clear anatomical boundaries and were easily dissected.

### Luciferase Reporter HEK cell transformation and assay conditions

HEK293 cells were grown routinely in DMEM + 10% foetal bovine serum and penicillin/streptomycin (all Gibco from Thermo-Fisher/Invitrogen). White-sided, clear-bottomed 96 well plates (Corning) were seeded at 8,000 cells/well in 100 µl/well medium without antibiotics. All transfections were carried out with Transfectin (Bio-Rad) at a ratio of 3/1 (0.3 µl Transfectin/ 100 ng DNA). Competition assays were carried out using VP16-TCF4 as the constitutive TCF (**Ewan et al., 2010**). The following plasmid amounts were used: VP16-TCF4: 43.5 ng/well; 25 ng/well reporter plasmids; 25 ng/well TCF isoform +pcDNA; 6.5 ng/well pcDNA-lacZ totalling 100 ng/well DNA. Transfectin was pipetted into 12.5 µl/well OptiMEM (Gibco/Thermo-Fisher) and the appropriate amount of DNA (above) into another 12.5 µl/well OptiMEM. The two were mixed and complexes were allowed to form over 20 min. The complexes were applied to the cells for 4 hr after which the medium was removed and replaced with a fresh 100 µl/well medium. After 2 days incubation, the medium was removed the cells were lysed in 50 µl/well GLO Lysis buffer (Promega) over 20 min on slow shake. The lysate was then split into two 25 µl/well fractions. The luciferase assay was carried out with 25 µl/well Bright-GLO (Promega) and the lacZ transfection control assay with 25 µl/well Beta-GLO (Promega). All measurements were taken on a Fluostar plate reader (BMG) set to luminescence mode.

### HEK293 cell immunohistochemistry and co-immunoprecipitation protocol

For immuno-histochemistry, following cell fixation and permeabilisation, samples were washed with PBS, blocked in 5% albumin, and then incubated at 4°C overnight in mouse anti-MYC antibody high-affinity 9E10 (SCBT) diluted 1:1000 in 1% albumin in PBS. Cells were washed with PBS and incubated at room temperature for 1 hr in Alexa Fluor 568 goat anti-mouse (Molecular Probes, Eugene, OR) diluted 1:1000 in 1% albumin in PBS. Samples were washed with PBS, incubated in DAPI (1:50,000) for 5 min at room temperature, washed again, and mounted.

For Co-IP, following transformation, 350 µl of Lysis Buffer 1 (below) was added and cells were resuspended off the plate by pipetting and lysed on ice for 1 hr gently mixing every 10 min. Samples were then centrifuged at 16.000 g for 15 min at 4°C, to enable removal of the supernatant and samples were then stored on ice. 70 µl of Lysis Buffer two was added to the pellet which was then sonicated three times for 5 cycles at 40% output (Branson 450 sonicator). Tubes were centrifuged at 16.000 g for 15 min at 4°C and the supernatant was collected and added to the supernatant of the previous centrifugation step. Proteins were quantified by BCA method (Sigma, BCA1).

Lysis Buffer 1: NaCl 150 mM, Tris-HCl 80mM pH7.2, NP-40 0.5%, Glycerol 20%. 10 µl of complete protease Inhibitor (Sigma, P-8340), 10 µl of phosphatase Inhibitor 1 (Sigma, P-2850)

and 10 µl of phosphatase Inhibitor 2 (Sigma, P-5726) were added before protein extraction for each 1 ml of lysis buffer.

Lysis Buffer 2: NaCl 300 mM, Tris-HCl 20mM pH7.2, SDS 0.01%, Triton 1%, EDTA 2 mM, add 10 µl of complete protease Inhibitor (Sigma, P-8340), 10 µl of phosphatase Inhibitor 1 (Sigma, P-2850) and 10 µl of phosphatase Inhibitor 2 (Sigma, P-5726) were added before protein extraction for each 1 ml of lysis buffer.

For each Immunoprecipitation (IP) experiment condition 40 µl of anti-myc bead slurry (Sigma, A7470) was added to 500 µl of protein sample diluted to 1 µg/µl in IPP buffer (Lysis Buffer 1 but with 5 mM EDTA and no Glycerol), and incubated for 4 hr at 4°C in a small rotator mixer at 4 RPM. Beads were spun for 30 s in a top bench centrifuge, the supernatant removed, and 500 µl of IPP buffer added, and rinsed three times. Samples were centrifuged at 12000 g for 1 min at 4°C, the supernatant was removed and 40 µl of 1.5x Laemmli Buffer was added. Proteins were eluted from the beads by incubating at 95°C for 5 min.

## Yeast Transformation

Fresh Y2Gold yeast colony was used to inoculate 3 ml of Yeast Peptone Dextrose Adenine (YPDA) liquid media for 8 hr and then 50 µl of the culture was used to inoculate 50 ml of YPDA and grow over night. The culture was grown until $OD^{600}$ was 0.16 and then centrifuged at 700 g for 5 min at room temperature. The supernatant was discarded and the yeast resuspended in 100 ml of fresh YPDA liquid media. The culture was grown for 3 hr or until $OD^{600}$ reached 0.4–0.5, and centrifuged at 700 g for 5 min at room temperature, supernatant was discarded and the yeast pellet was resuspended in 60 ml of water. Yeast were then centrifuged at 700 g for 5 min at room temperature, supernatant was discarded and yeast were resuspended in 3 ml of 1.1X TE/LiAc (Tris-HCl 1 mM, EDTA 1.1 mM, Li Acetate 110 mM, pH 7.5), centrifuged at 16000 g for 15 s, supernatant was discarded and yeast were resuspended in 1.2 ml of 1.1 TE/LiAc. Competent yeast were kept at room temperature and used within less than an hour.

For yeast transformation, herring testis DNA (10 µg/µl, Sigma, D-6898) was denatured at 100°C for 5 min and then chilled on ice. Cells were transformed by mixing 50 µg of herring testis DNA, 400 ng of bait and prey DNA (80 ng/µl), 50 µl of competent yeast, 500 µl of PEG/LiAc (40% Polyethilene glycol (Sigma, P-3640), Lithium Acetate 100 mM), vortexing and incubating at 30°C for 30 min mixing every 10 min. After this 20 µl of DMSO were added, cells were mixed by inversion, and heat shocked at 42°C for 15 min, mixing every 5 min. Cells were then chilled on ice for 2 min, centrifuged at 16000 g for 10 s, supernatant was discarded and yeast were resuspended in 200 µl of sterile TE 1X. All the yeast were plated on specific dropout selective agar plates (Materials and methods).

