## [Decision Letter]

**Acceptance summary:**

The authors identify and characterize the function of *tcf7l2* splice variants in zebrafish. They find that inclusion of a previously unidentified exon 5 is necessary and sufficient to confer transcriptional repression activity during neurectoderm patterning. Their data further show that the polypeptide encoded by exon 5 interacts with Tle corepressors in a phosphorylation-dependent manner, and suggest that this interaction mediates a general repressor function for Tcf7l2 that can be regulated in vivo. The authors propose that alternative splicing of *tcf7l2* exon 5 in zebrafish, and partially homologous exons in other species including humans, can influence the balance between activation and inhibition of Wnt pathway activity.

**Decision letter after peer review:**

Thank you for submitting your work entitled "Developmentally regulated *tcf7l2* splice variants mediate transcriptional repressor functions during eye formation" for consideration by *eLife*. Your article has been reviewed by two peer reviewers, and the evaluation has been overseen by a Reviewing Editor and a Senior Editor. The following individual involved in review of your submission has agreed to reveal their identity: Richard Dorsky (Reviewer #1).

There was a consensus among the reviewers that the manuscript addresses an important question of the mechanism of Tcf7l2 regulation of gene expression and provide an interesting data and model on the role of exon5 in conferring repressor activities to Tcf7l2, and its regulation by phosphorylation of residues encoded by exon5. However, both reviews noted similar weaknesses, especially that additional evidence for the repressor function should be provided, and relevance of the newly identified splice variant of tcfl72 has not been fully demonstrated in vivo. Moreover, the proposed role of Exon5 containing Tcf7l2 protein phosphorylation in promoting its repressor activity requires further experimental support. It was also felt that the amount of experimental effort needed to address these and other concerns raised by the reviewers would take longer than allowed for a revision. Therefore, it is with regret that I need to inform you that we cannot invite a revised manuscript at this time.

Reviewer #1:

In this manuscript by Young, et al., the authors identify specific splice variants of *tcf7l2* in zebrafish that include a novel exon and encode isoforms with repressor activity. They show that these variants are expressed in the developing forebrain, and are necessary and sufficient to promote eye formation along with *tcf7l1a/b*. The authors conclude that Tcf7l2 function in vivo can thus be modified by inclusion of this exon, affecting the balance between activation and repression of Wnt targets.

While the general concept of alternative transcripts affecting Tcf7l2 activation vs. repression has already been explored (Vacik, Stubbs and Lemke, 2011), the work presented here is novel, convincing, and carefully executed. I believe the current findings will be of interest to the field, considering the important roles of Tcf7l2 in development and disease. However the study has several weaker areas that should be addressed to provide a more complete story:

1) Most importantly, the function of protein products including exon 5 need to be more fully characterized. While a compelling case is made for their increased repressor activity, these experiments were carried out in the presumed absence of high Wnt or ß-catenin activity. It is not clear whether exon 5-containing products can also activate targets efficiently in the presence of ß-catenin, a function important for understanding the role of Tcf7l2 isoforms in vivo. For example, Tcf7l1 proteins can function as activators in vitro, but no obvious in vivo role for this activity has been revealed by genetic loss-of-function experiments.

2) Related to the first point, it would be helpful to know whether repressor-specific splice variants are primarily expressed in regions or tissues with low Wnt/ß-catenin signaling. This would support their potential role in target gene repression, and provide a direction for future studies exploring regulation of alternative splicing.

3) The role of Tcf7l2 co-repressor binding domain phosphorylation is incompletely explored. To establish the relevance of their binding and functional assays, the authors should provide evidence that these phosphorylation events take place in vivo, ideally in some regulated manner.

4) It would be helpful if the authors could provide a more comprehensive model for how they think Tcf7l2 contributes to A/P neural patterning. In addition to the partially redundant functions of zygotically expressed Tcf7l1a and Tcf7l1b in zebrafish, maternally deposited *tcf7l1a* mRNA plays an important role. It appears that the most significant factor in A/P patterning is therefore the gross amount of repressor Tcf isoforms relative to Wnt activity. Do the authors believe that exon 5-containing Tcf7l2 isoforms further contribute to this balance, or are other models possible?

Reviewer #2:

The manuscript by Young and colleagues addresses the question of how the function of Tcf protein Tcf7l2 is altered as a result of alternative splicing.

To understand the contribution of different *tcf7l2* splice variants to protein function the authors focused on the CDRD region of the protein, which is known to be alternatively spliced and likely to influence the transcriptional function of the protein. They found that in zebrafish, this region is encoded in part by a previously un-described exon 5. During early somitogenesis stages (12 hpf), two splice variants are expressed, one that includes exon 4 but not 5 (*4L-tcf7l2*) and one that includes both exons 4 and 5 (*45L-tcf7l2*). By in situ hybridization the authors show that *tcf7l2* is expressed throughout most of the forebrain, including the eye field, from late gastrula stage, and by analyzing an available mutant they find that *tcf7l2* homozygous mutant embryos have smaller eyes than normal. By misexpression experiments the authors show that only the protein encoded by *45L-tcf7l2* can rescue the eyeless phenotype of *tcf7l1a^-/-^/tcf7l1b* morphant embryos and that in vitro this protein represses transcription of Wnt target genes and appears to have a higher affinity to TLE. Finally the authors propose that two amino acid encoded by exon 5 are likely phosphorylated and this phosphorylation could be critical to the repressor activity of the 45L-Tcf7l2 variant.

Overall the manuscript addresses an important question in the context of a protein that is implicated in developmental processes as well as in cancer and type-2 diabetes. The manuscript is clearly written and the data are generally of high quality. However, there are several major concerns that weaken the overall strength of the work:

1) The authors propose that 45L-Tcf7l2 is the relevant form for repression of Wnt target genes during eye development. If so, then specific loss of function of this form (using the splice MO) should lead to at least the same eye phenotype as seen in *tcf7l2* mutants. This is not the case, and the authors discuss the effects of the splicing MO only in the context of *tcf7l1a^-/-^* embryos.

2) The conclusion that 45L-Tcf7l2 functions as a repressor relies on its ability to interfere with the strong activation of luciferase by a Tcf7l2-VP16 chimeric protein. This does not prove that 45L-Tcf7l2 acts as a transcriptional repressor, and given that the transcriptional activity of Tcf7l2 variants is a key issue in this work, a more direct proof of repressor activity should be provided.

3) The co-immunoprecipitation experiments are aimed at addressing the differences in binding affinity of Tcf7l2 variants to TLE or β-Catenin. This is problematic given that co-IP is not a quantitative assay, hence it cannot fully support the conclusion of different affinities.

4) The authors propose that the differences between repressor activities of 4L-Tcf7l2 and 45L-Tcf7l2 are caused by phosphorylation of two amino acids in the region encoded by exon 5, and address this by mutating the two residues to Alanine. However, this is very circumstantial and no direct proof of phosphorylation is provided.

In summary, although the work proposes an important mechanism for regulating transcriptional activity of a TCF factor, I feel that the conclusions are not as strongly supported as they need to be for publication in a high-profile journal such as *eLife* and the contribution of the work is currently incremental.

[Editors’ note: what now follows is the decision letter after the authors submitted for further consideration.]

Thank you for resubmitting your work entitled "Developmentally regulated *tcf7l2* splice variants mediate transcriptional repressor functions during eye formation" for further consideration by *eLife*. Your revised article has been evaluated by Didier Stainier as the Senior Editor, a Reviewing Editor and two reviewers.

The manuscript has been improved but there are some remaining issues that need to be addressed before acceptance, as outlined below:

In this manuscript by Young, et al., the authors identify and characterize the function of *tcf7l2* splice variants in zebrafish. They find that inclusion of a previously unidentified exon, which they call exon 5, is necessary and sufficient to confer transcriptional repression activity during neurectoderm patterning. Their data further show that the polypeptide encoded by exon 5 interacts with Tle corepressors in a phosphorylation-dependent manner, and suggest that this interaction mediates a general repressor function for Tcf7l2 that can be regulated in vivo. The authors propose that alternative splicing of *tcf7l2* exon 5 in zebrafish, and partially homologous exons in other species including humans, can influence the balance between activation and inhibition of Wnt pathway activity.

This is an extremely comprehensive study, including diverse genetic and biochemical approaches, that makes a clear and convincing case for the mechanistic regulation of a gene important for development and disease. The data are novel and are likely to be of interest to a wide audience. While similar mechanisms have been described for other Lef/Tcf family members, they have not been tested as rigorously.

In the revised version the authors have addressed several weak points that were present in the initial submission and improved the manuscript. The additional reporter assay for repressor function is more convincing and the identification of phosphorylated residues by mass spectrometry and their mutational analysis provides strong evidence to support the proposed mechanism that regulates the repressor function of Tcf7l2. The authors have also added data supporting relevance of their findings to human TCF proteins. There are only a few areas where the manuscript could be strengthened:

1) Inclusion of exon 5 is necessary to rescue the eyeless phenotype following loss of *tcf7l1a/b*, and the authors hypothesize that this is due to the increased affinity for Tle3 from phosphorylated residues in this isoform. However, the 4L and AA isoforms still retain some Tle3 binding affinity, and as pointed out in the Discussion the balance between activation and repression could depend on many factors. To more fully support their model, it seems like a relatively simple experiment would be to overexpress *tle3* mRNA with the non-rescuing isoforms and test whether this changes their activity.

Whereas the above experiments are not considered as essential for the manuscript's acceptance, it would strengthen the model. Minimally, a fuller discussion of the concept that "the balance between the levels of Tcf transcriptional repressing versus activating variants" is required. In this discussion, it would be important to mention the observation from data not shown that expression of the 45L form in wild-type background did not cause an overt phenotype (subsection “Zebrafish *tcf7l2* exon 5 and human *tcf7l2* exon 3a containing variants are able to restore eye formation upon loss of *tcf7l1a/b* function”, first paragraph). Clearly, this balance might become critical only in certain genotypic scenarios.

2) The authors repeatedly describe *tcf7l2* expression as being in the anterior neural ectoderm or in the forebrain. However, they do not distinguish between presumptive telencephalon and diencephalon. Both Figure 1F/G and images from Young et al., 2002, suggest that expression is primarily diencephalic, and while it may partially overlap with *tcf7l1a/b* it is not identical to these genes. These subtle differences in expression may help explain non-redundant functions of individual Lef/Tcf family members, in addition to their ability to act as transcriptional activators or repressors.

3) The reviewers think that Supplementary Figure 3 should be a main figure rather than supplemental.

---

## [Author Response]

[Editors’ note: the author responses to the first round of peer review follow.]

There was a consensus among the reviewers that the manuscript addresses an important question of the mechanism of Tcf7l2 regulation of gene expression and provide an interesting data and model on the role of exon5 in conferring repressor activities to Tcf7l2, and its regulation by phosphorylation of residues encoded by exon5. However, both reviews noted similar weaknesses, especially that additional evidence for the repressor function should be provided, and relevance of the newly identified splice variant of tcfl72 has not been fully demonstrated in vivo. Moreover, the proposed role of Exon5 containing Tcf7l2 protein phosphorylation in promoting its repressor activity requires further experimental support. It was also felt that the amount of experimental effort needed to address these and other concerns raised by the reviewers would take longer than allowed for a revision. Therefore, it is with regret that I need to inform you that we cannot invite a revised manuscript at this time.

The major changes between the original version of the manuscript and the new version are that we now show that exon5 coded Tcf7l2 variants can repress the luciferase reporter under high β-catenin activity conditions and add mass spectrometry evidence that amino acids in the exon5 coded region of Tcf7l2 are phosphorylated. We show that phosphomimicking amino-acids enable exon5 coded Tcf7l2 variants to retain full activity whereas non-phosphorylatable amino acids do not. Finally, we add new data showing that alternative exons in human Tcf7l2 have comparable activity to those in zebrafish suggesting evolutionary conservation of the role of splicing in mediating Tcf7l2 function. Below, we provide a more detailed description of our responses to the reviewer comments and the new data that is included in the paper.

We have addressed all the comments from reviewer 1 and three out of four major comments from reviewer 2. Below we describe the experiments that we have added to the manuscript and how they address the reviewers’ comments, and explain why one of the comments could not be addressed.

Reviewer #1:[…] 1) Most importantly, the function of protein products including exon 5 need to be more fully characterized. While a compelling case is made for their increased repressor activity, these experiments were carried out in the presumed absence of high Wnt or ß-catenin activity. It is not clear whether exon 5-containing products can also activate targets efficiently in the presence of ß-catenin, a function important for understanding the role of Tcf7l2 isoforms in vivo. For example, Tcf7l1 proteins can function as activators in vitro, but no obvious in vivo role for this activity has been revealed by genetic loss-of-function experiments.

We have added a new dataset of luciferase reporter assays in conditions of high β-catenin activity. *Tcf7l2* variants were co-expressed with the Flag-Ax2 construct that outcompetes the endogenous GSK-3β such that it cannot phosphorylate βcatenin, leading to an increase in the levels of β-catenin protein and concomitant downstream pathway activity (manuscript Figure 3A-F). In this experimental setup, exon 5 coding Tcf7l2 variants show less transactivation capacity compared to the variants that do not include exon5 on all the gene promotors we tested. This suggests that inclusion of the exon5 coded fragment in Tcf7l2 promotes its repressive function. We hope that these additional experiments are sufficient to address concerns raised by reviewer 1 (comment #1) and reviewer 2 (comment #2).

2) Related to the first point, it would be helpful to know whether repressor-specific splice variants are primarily expressed in regions or tissues with low Wnt/ß-catenin signaling. This would support their potential role in target gene repression, and provide a direction for future studies exploring regulation of alternative splicing.

We thank the reviewer 1 for pointing this out and agree with this suggestion. We have added additional text explaining how we envisage Tcf7l2 works in the context of Wnt and other Tcfs to contribute to AP neural patterning (subsection “Tcf7l2 transcriptional repression function contributes to forebrain patterning”, last paragraph).

3) The role of Tcf7l2 co-repressor binding domain phosphorylation is incompletely explored. To establish the relevance of their binding and functional assays, the authors should provide evidence that these phosphorylation events take place in vivo, ideally in some regulated manner.

Both reviewer 1 (comment #3) and reviewer 2 (comment #4) comment on the lack of direct evidence for the proposed phosphorylation in the Tcf7l2 exon5 coded region. To address this point, we have added the results from a mass spec experiment on pulled-down Tcf7l2 containing exon 5 expressed in HEK cells. We find that the trypsin fragment containing the region coded by *tcf7l2* exon5 is phosphorylated. This result confirms phosphorylation of the amino acids we have studied in the region of Tcf7l2 coded by exon 5.

4) It would be helpful if the authors could provide a more comprehensive model for how they think Tcf7l2 contributes to A/P neural patterning. In addition to the partially redundant functions of zygotically expressed Tcf7l1a and Tcf7l1b in zebrafish, maternally deposited tcf7l1a mRNA plays an important role. It appears that the most significant factor in A/P patterning is therefore the gross amount of repressor Tcf isoforms relative to Wnt activity. Do the authors believe that exon 5-containing Tcf7l2 isoforms further contribute to this balance, or are other models possible?

We thank the reviewer 1 for pointing this out and agree with this suggestion. We have added additional text explaining how we envisage Tcf7l2 works in the context of Wnt and other Tcfs to contribute to AP neural patterning (subsection “Tcf7l2 transcriptional repression function contributes to forebrain patterning”, last paragraph).

Reviewer #2:[…] 1) The authors propose that 45L-Tcf7l2 is the relevant form for repression of Wnt target genes during eye development. If so, then specific loss of function of this form (using the splice MO) should lead to at least the same eye phenotype as seen in tcf7l2 mutants. This is not the case, and the authors discuss the effects of the splicing MO only in the context of tcf7l1a^-/-^ embryos.

As the reviewer predicts, injecting the *tcf7l2* exon 5 specific splicing MO in wildtype embryos leads to reduced eye size. This result implies that exon5 coded stretch of the protein mediates the repressor function of Tcf7l2 and that consequently loss of this exon has a comparable effect on eye formation as observed due to reduced pathway repression seen in the *tcf7l1a* mutant condition. We have added this data to Figure 3—figure supplement 2C-E and elaborate on the interpretation of this result in the Discussion.

2) The conclusion that 45L-Tcf7l2 functions as a repressor relies on its ability to interfere with the strong activation of luciferase by a Tcf7l2-VP16 chimeric protein. This does not prove that 45L-Tcf7l2 acts as a transcriptional repressor, and given that the transcriptional activity of Tcf7l2 variants is a key issue in this work, a more direct proof of repressor activity should be provided.

In comment #2, reviewer 2 asked us to assess if *tcf7l2* variants containing exon 5 were expressed in regions of low levels of Wnt activity. We agree that assessing the expression pattern of alternative exons would be a great contribution to this manuscript. However, given that exon 5 is only 60nt long, detecting this RNA fragment by in situ hybridisation is a technical challenge. We tried to detect exon 5 specific containing splice variants by in situ hybridisation using a sensitive fluorescent multiplexed technique (Molecular Instruments). However, we could not detect any signal over background. We think that the best way to address this issue may be by raising antibodies specific to exon 5-forms, a task that would be beyond the scope of this manuscript.

Please see our response to reviewer 1 comment #1.

3) The co-immunoprecipitation experiments are aimed at addressing the differences in binding affinity of Tcf7l2 variants to TLE or β-Catenin. This is problematic given that co-IP is not a quantitative assay, hence it cannot fully support the conclusion of different affinities.

We also thank reviewer 2 for this suggestion, and based on this we have toned down the interpretation of the interaction we see between Tcf7l2 exon 5 variants and Tle3b in the coIP experiments (subsection “Inclusion of exon 5 enhances the interaction between Tcf7l2 and Tle3b”, last paragraph).

4) The authors propose that the differences between repressor activities of 4L-Tcf7l2 and 45L-Tcf7l2 are caused by phosphorylation of two amino acids in the region encoded by exon 5, and address this by mutating the two residues to Alanine. However, this is very circumstantial and no direct proof of phosphorylation is provided.

Please see our response to reviewer 1 comment #3.

[Editors' note: the author responses to the re-review follow.]

The manuscript has been improved but there are some remaining issues that need to be addressed before acceptance, as outlined below:[…] 1) Inclusion of exon 5 is necessary to rescue the eyeless phenotype following loss of tcf7l1a/b, and the authors hypothesize that this is due to the increased affinity for Tle3 from phosphorylated residues in this isoform. However, the 4L and AA isoforms still retain some Tle3 binding affinity, and as pointed out in the Discussion the balance between activation and repression could depend on many factors. To more fully support their model, it seems like a relatively simple experiment would be to overexpress tle3 mRNA with the non-rescuing isoforms and test whether this changes their activity.

We agree that the results from the experiment suggested by the reviewers would add relevant information to the manuscript and performed it. However, overexpression of both 10pg of *tcf7l2-AA* and 10pg of *tle3b* mRNA did not rescue eye formation in *tcf7l1a^-/-^tcf7l1b* morphants. Moreover, overexpression of 10pg of *tle3b* mRNA in *tcf7l1a^-/-^* mutants led to eyeless embryos. We did not explore why this phenotype arose but we speculate that it may be due to *tle3b* together with *tcf7l1a* having an impact on mesoderm or ectoderm patterning prior to eye formation. As this result does not inform our study, we do not include it in the manuscript.

Whereas the above experiments are not considered as essential for the manuscript's acceptance, it would strengthen the model. Minimally, a fuller discussion of the concept that "the balance between the levels of Tcf transcriptional repressing versus activating variants" is required. In this discussion, it would be important to mention the observation from data not shown that expression of the 45L form in wild-type background did not cause an overt phenotype (subsection “Zebrafish tcf7l2 exon 5 and human tcf7l2 exon 3a containing variants are able to restore eye formation upon loss of tcf7l1a/b function”, first paragraph). Clearly, this balance might become critical only in certain genotypic scenarios.

We have further elaborated on balance between activating and repressing variants in the Discussion, subsections “Tcf7l2 transcriptional repression function contributes to forebrain patterning”, second paragraph and “Phosphorylation contributes to the balance of transcriptional activation-repression by mediating the function of Tcf7l2 as a transcriptional repressor”, first paragraph.

2) The authors repeatedly describe tcf7l2 expression as being in the anterior neural ectoderm or in the forebrain. However, they do not distinguish between presumptive telencephalon and diencephalon. Both Figure 1F/G and images from Young et al., 2002, suggest that expression is primarily diencephalic, and while it may partially overlap with tcf7l1a/b it is not identical to these genes. These subtle differences in expression may help explain non-redundant functions of individual Lef/Tcf family members, in addition to their ability to act as transcriptional activators or repressors.

We agree with the reviewers and have further elaborated on the expression pattern of *tcf7l2* in the anterior neuroectoderm (subsection “*tcf7l2* is broadly expressed in the anterior neural plate”, last paragraph) and have also included details on how the differences in the expression between *tcf7l2* and *tcf7l1a/b* could lead to functional specificity to the Discussion (first paragraph).

3) The reviewers think that Supplementary Figure 3 should be a main figure rather than supplemental.

We agree with the reviewers and Supplementary Figure 3 is now main Figure 2.